# L-RNA aptamer-based CXCL12 inhibition combined with radiotherapy and bevacizumab in newly-diagnosed glioblastoma: expansion of the phase I/II GLORIA trial

Rapid vascular recovery is a key feature preceding glioblastoma (GBM) recurrence after radiotherapy (RT). We performed spatial expression analyses, providing a rationale for dual inhibition of two non-redundant, spatially distinct acting factors, CXCL12 and VEGF. Subsequently, we expanded a multicentric phase 1/2 trial (NCT04121455), which initially combined RT and the CXCL12-neutralizing L-RNA-aptamer olaptesed pegol (NOX-A12) in patients with incompletely resected, newly-diagnosed GBM lacking *MGMT* promoter methylation. The primary endpoint was safety, secondary endpoints included maximum tolerable dose, recommended phase 2 dose, NOX-A12 plasma levels, topography of recurrence, tumor vascularization, neurologic assessment in neuro-oncology (NANO), quality of life, median progression-free survival (PFS), 6-months PFS and overall survival (OS). For the expansion arm, six patients were included that additionally received the VEGF-targeting antibody bevacizumab (BEV) to RT and NOX-A12. Combinatory treatment was well-tolerated and safe with no treatment-related deaths, resulting in abrogated tumor perfusion (rCBV, FTB$^{high}$) and delayed tumor regrowth as per mRANO. Median progression-free (PFS) and overall survival (OS) after RT + BEV + NOX-A12 were 9.1 and 19.9 months, respectively, significantly outperforming RT + NOX-A12 ($p = 0.009$; $p = 0.021$) in a post-hoc comparative analysis, with two patients exceeding 2-year OS. These findings establish proof-of-principle for dual inhibition of CXCL12 and VEGF in patients with newly-diagnosed GBM following RT.

Glioblastoma (GBM) is the most common malignant primary brain tumor in adults and remains associated with a severely limited prognosis[1]. Standard-of-care (SOC) treatment consists of maximum-feasible resection, external-beam radiotherapy (RT) and adjuvant chemotherapy with temozolomide (TMZ)[2,3]. Roughly 60%

of the tumors harbor an unmethylated O$^6$-methylguanine DNA methyltransferase (*MGMT*) promoter[4], which confers resistance to alkylating chemotherapy. These patients face a dismal median progression-free survival (PFS) of 4–6 months and a median overall survival (OS) of 10-15 months[5–8]. Complete tumor resection is an

✉ e-mail: Frank.Giordano@umm.de; michael.hoelzel@ukbonn.de

independent prognostic factor[9,10] with a significant impact on OS[11,12].

A key feature of GBM recurrence is the efficient modulation of the tumor microenvironment (TME) to restore the microvasculature[13], which provides residual tumor cells with vital nutrients and oxygen by exploiting two distinct mechanistic pathways, *angiogenesis* and *vasculogenesis*.

Angiogenesis is characterized by hypoxia-induced autocrine and paracrine synthesis of vascular endothelial growth factor A (VEGFA)[14], which mediates local sprouting of pre-existing vessels[15,16]. Following promising preclinical data indicating a synergistic effect with RT[17], the VEGFA-targeting monoclonal antibody bevacizumab (BEV) has been previously investigated as a potential therapeutic agent in GBM and yielded improved response rates, prolonged PFS and reduced glucocorticoid requirements in recurrent GBM[18,19]. However, while providing significantly improved PFS, its addition to chemoradiotherapy repeatedly failed to demonstrate an OS benefit in phase 2[20] and phase 3[21,22] trials.

In contrast, vasculogenesis describes de novo vessel formation mediated by bone marrow-derived progenitor cells (BMDCs)[23–27]. Radiation-induced loss of endothelial cells, accumulation of reactive oxygen species, and aggressive tumor growth with aberrant neovascularization collectively lead to impaired tumor perfusion, triggering TGF-β- augmented induction of the hypoxia-related factor HIF1α. This, in turn, attracts CXCR4$^+$ BMDCs along a gradient of the CXCR4 ligand CXCL12[24,28,29]. Elevated CXCL12 expression facilitates not only BMDC migration but also their adhesion and homing to hypoxic tumor regions[24]. CXCL12 has furthermore been reported to repel and sequester T cells[30–32], promote GBM cell invasion and decrease apoptosis[33,34]. Pre-clinical studies have demonstrated improved intracranial tumor control in orthotopic GBM models after RT and subsequent treatment with the CXCR4-inhibiting bicyclam derivative plerixafor[35]. In a phase 1/2 trial with 20 high-grade glioma patients, plerixafor administration was shown to be safe and yielded favorable clinical outcomes. In this trial, the unusually frequent occurrence of out-of-field recurrences suggests a pivotal role of CXCR4$^+$ cells in locally confined post-radiogenic neovascularization[36].

Combined treatment with RT and the CXCL12-inhibiting pegylated L-RNA aptamer olaptesed pegol (NOX-A12) demonstrated significant tumor reduction and sustained complete regressions in an autochthonous rat brain tumor model mimicking highly treatment-refractory GBM[37]. We previously reported on safety and subgroup-definable clinical efficacy for RT and NOX-A12 in newly-diagnosed, *MGMT*-unmethylated and incompletely resected GBM[38]. In this dose-escalation part of the multicentric GLORIA trial (NCT04121455), a higher frequency of CXCL12$^+$ endothelial and glioma cells (EG12) was significantly associated with prolonged PFS under NOX-A12, but not in a SOC cohort. GLORIA patients with EG12$^{high}$ GBM derived the greatest benefit from treatment. The inferior outcome of EG12$^{low}$ GBM and the lack of translation into a significant OS benefit raise questions about the individual TME composition and factors driving drug resistance, both at baseline and recurrence. To date, no clinical trial has investigated dual targeting of CXCL12 and VEGF as two key mediators of post-radiogenic revascularization in GBM.

We here report on a spatially resolved mechanistic concept of GBM vascularization, providing a rationale for a combinatorial first-line treatment with RT plus dual inhibition of post-radiogenic angiogenesis and vasculogenesis, demonstrating early signs of improved clinical efficacy for this approach in the multicentric phase 1/2 GLORIA trial.

## Results

### Characterization of compartment-specific vascularization patterns

In a stepwise approach with increasing spatial resolution (Fig. 1a), we first aimed to investigate the relationship between hypoxia, VEGFA and CXCL12 expression in GBM using the publicly available bulk RNA sequencing dataset ($n = 159$) from The Cancer Genome Atlas (TCGA)[39]. *VEGFA* expression was positively correlated ($r_s = 0.59$, $p < 0.001$) with the Winter hypoxia score[40], a widely used hypoxia gene signature, but this was not the case for *CXCL12* ($r_s = -0.07$, $p = 0.358$). Carboanhydrase 9 (CA9), a hypoxia-inducible gene and well-established marker for detection of hypoxic regions by immunohistochemistry[41], also showed a strong positive correlation with *VEGFA* ($r_s = 0.715$, $p < 0.001$), but not *CXCL12* ($r_s = -0.16$; $p = 0.043$; Fig. 1b).

Next, we assessed regional mRNA expression patterns of *VEGFA* and *CXCL12* in human GBM samples ($n = 122$) from the Ivy Glioblastoma Atlas Project[42] with available neuropathology-based tissue mapping (Fig. 1c). *CXCL12* was significantly higher expressed than both *VEGFA* and *CA9* in the infiltration zone and hypervascularized tumor compartments ($p < 0.001$, Supplementary Fig. 1a), while *VEGFA* and *CA9* were particularly abundant in necrotic tumor areas ($p < 0.001$, Supplementary Fig. 1b). Confirming the RNA sequencing data from bulk GBM[39], *CA9* and *VEGFA* were significantly positively correlated ($r_s = 0.783$; $p < 0.001$), in contrast to *CXCL12* and *VEGFA* ($r_s = -0.474$; $p < 0.001$) (Supplementary Fig. 1c).

Aiming for single-cell spatial resolution, we performed multiplex-immunofluorescence (mIF) stainings on human GBM tissue samples ($n = 5$, Fig. 1d and Supplementary Fig. 2). Consistent with our previous findings[38], we detected the highest frequency of CXCL12 positivity in endothelial cells, particularly in the microvascular proliferation zone (Fig. 1e). The hypoxia marker CA9, which exhibited one of the strongest positive correlations with VEGFA in previous analyses, was primarily expressed around necrotic tumor areas. Although co-expression of CXCL12 and CA9 was observed, CXCL12 expression was largely independent of CA9 positivity (Fig. 1f). This finding supports the notion that CXCL12 is regulated not solely by hypoxia but also by additional mechanisms across diverse cellular sources. In summary, these findings suggest both shared and distinct regulatory mechanisms of vascularization in GBM, providing a strong rationale for dual therapeutic targeting of CXCL12 and VEGFA (Fig. 2).

To further test and validate this hypothesis, we next generated patient-derived glioblastoma organoids (GBOs) from surgical specimens cultured on human brain slices (Fig. 3a) and performed spatial transcriptomics on a representative model. The GBO tumor tissue displayed extensive hypoxic areas with pronounced VEGF expression at the core, whereas CXCL12 expression was sparse and, when present, colocalized predominantly with few endothelial cells, and, to a lesser extent, pericyte-, macrophage- and glioma-related transcripts (Fig. 3b). Notably, CXCL12 expression was detected in normal brain vasculature of the surrounding, the GBO embedding tissue (Fig. 3c). To overcome the mechanistical limitations of revascularization assessment in a non-perfused GBO model, we conducted parallel spatial transcriptomics analyses on a representative GBM patient specimen with well-defined zonal architecture (Fig. 3d). Consistent with our previously reported observations, CXCL12 expression localized to endothelial cells, pericytes, macrophages/microglia and glioma cells. In contrast, VEGF expression strongly correlated with hypoxia-related genes, while CXCL12 showed only a weak association with hypoxia but a robust link to the described cellular compartments (Fig. 3e).

Taken together with our preceding analyses, these findings further supported the spatially and mechanistically distinct involvement of both VEGF- and CXCL12-mediated pathways in GBM vascular remodeling. Consequently, within the GLORIA trial, we evaluated the potential additional clinical benefit of VEGFA-targeting with BEV in combination with NOX-A12.

### Trial design, enrollment and patient characteristics

The GLORIA trial (NCT04121455) is a multicentric phase 1/2 trial conducted at six major neuro-oncology centers in Germany. It was designed to evaluate the safety and efficacy of RT combined with

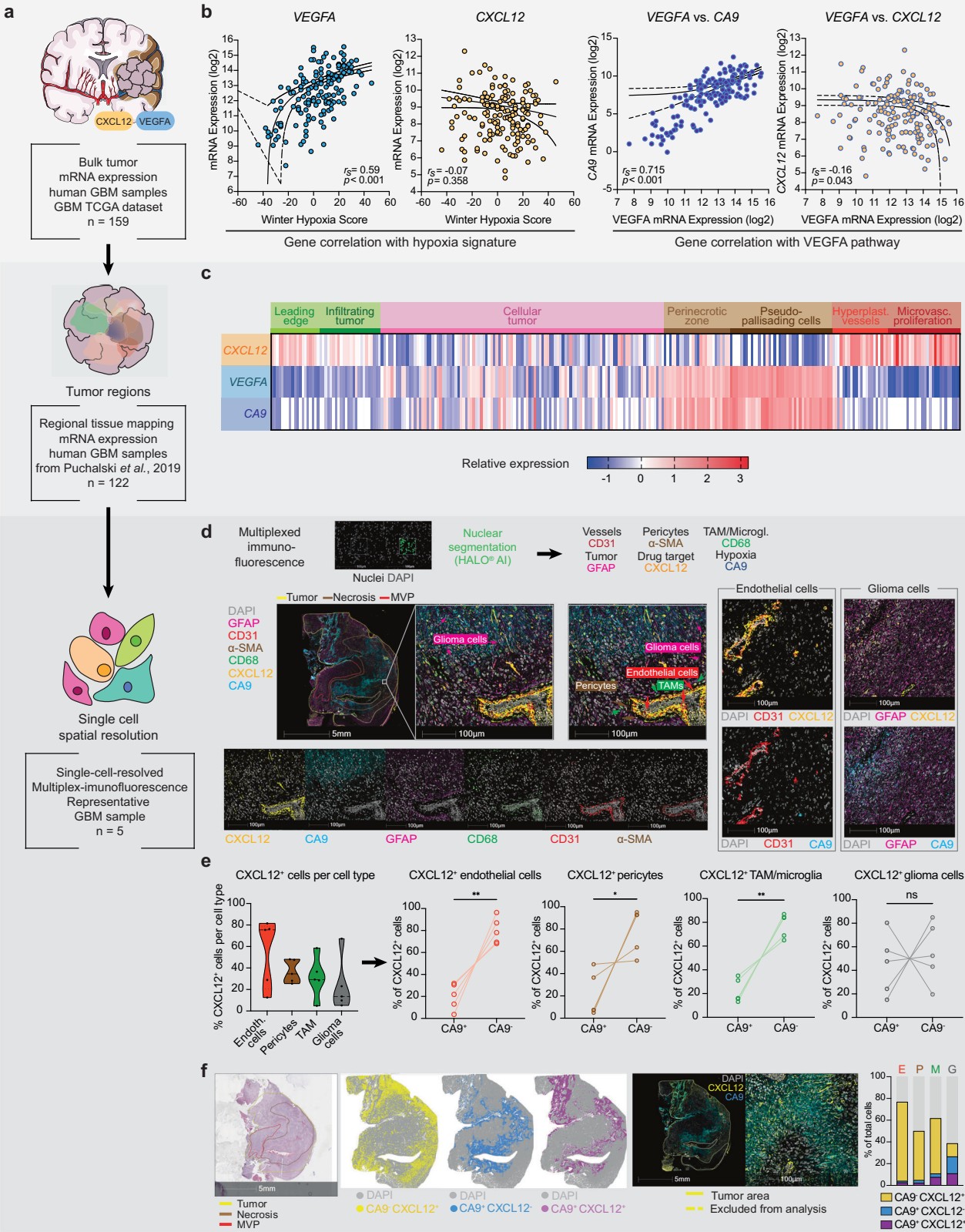

continuous intravenous (i.v). treatment with NOX-A12 in newly-diagnosed, incompletely resected IDH-wildtype GBM (CNS WHO grade 4) lacking *MGMT* promoter methylation (Fig. 4a). The primary endpoint was safety, secondary endpoints included maximum tolerable dose (MTD), recommended phase II dose (RP2D), NOX-A12 plasma levels, topography of recurrence, tumor vascularization, neurologic assessment in neuro-oncology (NANO), quality of life (QOL), median PFS,

6-months PFS and OS. In the dose escalation part of the trial, NOX-A12 was administered at escalating dose levels (DLs) of 200, 400 and 600 mg per week. Patients in the first expansion arm received 600 mg NOX-A12 per week combined with biweekly 10 mg/kg body weight BEV.

Between September 2019 and September 2021, three patients were enrolled at each DL of the dose escalation part of the trial

**Fig. 1 | Angio- and vasculogenesis are compartment-specific, non-redundant drivers of tumor vascularization in GBM. a** Graphical workflow. **b** Left: Spearman's rank correlation ($r_s$) between Winter Hypoxia Score and *VEGFA* (blue) or *CXCL12* (yellow) mRNA expression, n = 159 GBM cases. Right: Spearman's correlation ($r_s$) between *VEGFA* and *CA9* mRNA expression (dark blue) or *CXCL12* (yellow) mRNA expression, *n* = 159 GBM cases. Dataset from The Cancer Genome Atlas[39]. $r_s$- and *p*-values (two-tailed) are shown, dashed lines indicate 95% confidence intervals. **c** Heatmap illustrating relative compartment-specific expression of *CXCL12, VEGFA* and *CA9* with color gradient from low (dark blue) to high (dark red). Tumor compartments are indicated above. Dataset from Puchalski et al.[42] (*n* = 122 GBM cases). **d** Representative multiplex-immunofluorescence staining of GBM tissue (*n* = 5 patients). Upper panel: Workflow illustration and marker-based cell population definition for CXCL12 and CA9 positivity. Lower panels: Representative images of the transition zone between hypoxic and vascularized tumor regions showing CXCL12, CA9 and compartment-specific markers. Additional images demonstrate CXCL12 and CA9 colocalization with endothelial and glioma cells. **e** Frequency of CA9+ and CXCL12+ cells across cell types in GBM samples (*n* = 5 patients). Left: Violin plots depicting overall CXCL12 positivity. Right: Connected dot plots showing percentages of CA9+CXCL12+ (left) and CA9-CXCL12+ (right) cells in endothelial cells (red), pericytes (brown), TAM/microglia (green) and glioma cells (gray). Paired two-tailed *t* test; ns: not significant ($p > 0.05$); ** $p < 0.01$; ***$p < 0.001$. Exact p-values are provided in the Source Data. **f** Representative images of the same sample as in (**d**) using H&E and multiplex-immunofluorescence (*n* = 5 patients). Distribution of CA9-CXCL12+ (yellow), CA9+ CXCL12- (dark blue) and CA9+ CXCL12+ (purple) cells is shown using automated nuclear segmentation (DAPI, gray). Stacked bar plot depicts percentage of CA9- CXCL12+ (yellow), CA9+ CXCL12- (dark blue) and CA9+ CXCL12+ (purple) cells out of total endothelial cells (E), pericytes (P), macrophages/microglia (M) and glioma cells (G) in the same tissue (*n* = 1 patient). Source data are provided as a Source Data file. GBM: glioblastoma; MVP: microvascular proliferation; NOX-A12: olaptesed pegol; RT: radiotherapy; TAM: tumor-associated macrophages.

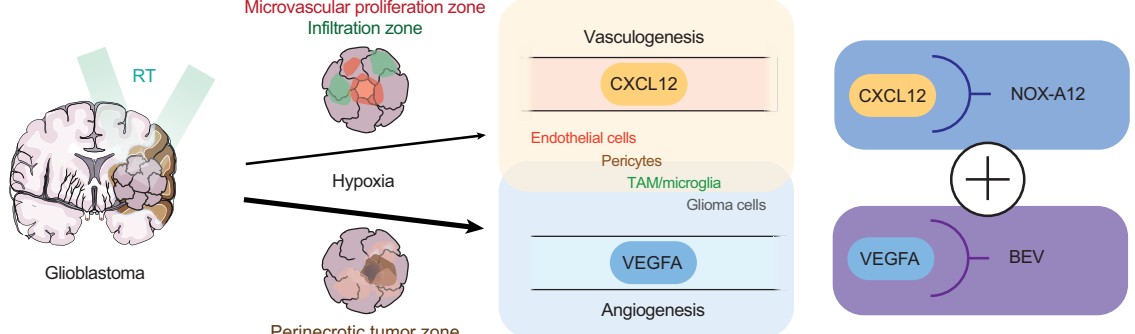

**Fig. 2 | Schematic illustration of the hypothesis.** BEV: bevacizumab; NOX-A12: olaptesed pegol; RT: radiotherapy.

(Supplementary Fig. 3) as previously reported[38]. One patient of DL 3 dropped out and was replaced, resulting in a total of ten patients treated with RT and NOX-A12. Between November 2021 and May 2022, six additional patients were enrolled in the expansion arm receiving RT, NOX-A12, and BEV. The median age at diagnosis of the full trial population (*n* = 16) was 63 (range 43 to 79) years. Of these, 14 patients had undergone partial resection; two were not amenable to resection and received biopsy only. No patient had undergone gross total resection. The median residual tumor volume at baseline was 6.2 (range 0.9–34.1) cm³ and the median highly-perfused tumor fraction (FTB^high) at baseline was 26.4% (range 1.8–67.5%). Additional baseline characteristics are shown in Fig. 4b, with a complete tabular summary provided in Supplementary Data 1. The median time to permanent discontinuation of the experimental drug was 5.4 (range 2.9–11.1) months for RT + NOX-A12 and 9.1 (4.1–25.7) months for RT + NOX-A12 + BEV.

## Pharmacokinetics

NOX-A12 plasma levels were not influenced by additional BEV treatment, reaching a stable steady-state after approximately one week in all patients and surpassing 1.5 μM, which was considered to be the minimum plasma level required for disrupting CXCL12-mediated migration while minimizing bone marrow cell mobilization[43]. With increasing NOX-A12 dosage, plasma levels increased correspondingly ($p = 0.007$), exceeding the stable CXCL12 levels ($p = 0.116$; Supplementary Fig. 4). NOX-A12 plasma levels were significantly higher from DL 1 to DL 2 ($p = 0.020$) and DL3 ($p = 0.020$), but did not differ significantly with additional BEV treatment ($p = 0.270$).

## Safety

The favorable safety profile of RT and NOX-A12[38] remained unchanged with the addition of BEV in the expansion arm (Fig. 5a). Triple therapy (RT + NOX-A12 + BEV) was safe and well tolerated, with no patient discontinuing treatment due to adverse events (AEs). No dose-limiting toxicities or treatment-related deaths were observed. Treatment was discontinued per protocol at the scheduled end of treatment (EOT) after 26 weeks in one patient. Five patients were treated beyond progression, and treatment was stopped as per the investigator's decision due to suspected decline in patient benefit or deterioration of general condition. Among 139 AEs reported in the expansion arm receiving RT + NOX-A12 + BEV, only two (1.5%) were considered solely related to NOX-A12 (Fig. 5b). Of all grade ≥ 2 AEs (*n* = 97), 4 grade 2 events (4.1%) were at least attributed to NOX-A12. The majority of AEs were of grade 1 or 2 (91.4%) and were either unrelated or associated to the tumor or RT. No grade 4 or 5 events occurred. The most common treatment-emergent AE (TEAEs, *n* = 139) was alopecia, reported in five patients with a maximum severity of grade 2 (Supplementary Fig. 5). Four patients experienced hypertension, likely related to BEV, which was effectively managed with pharmaceutical interventions (Supplementary Fig. 6). Equally, four patients each reported headache and fatigue. An increase in alanine aminotransferase, the most concerning treatment-related AE (TRAE) under RT + NOX-A12, occurred only once in the expansion arm (grade 2). Complete listings of all AEs, TEAEs and TRAEs for both RT + NOX-A12 and RT + NOX-A12 + BEV are available in Supplementary Data 1, Supplementary Data 2 and Supplementary Data 3, respectively. MTD and RP2D have been reported previously[38], with no additional dose-defining findings arising from the expansion arm.

## Radiographic and clinical response

All 16 patients enrolled were included in the response analysis (Fig. 6a). Among patients treated with NOX-A12 + BEV (*n* = 6), one patient achieved a confirmed complete response (CR), four partial response (PR) and one stable disease, resulting in an overall response rate of 83.3%. In addition, two patients (33.3%) surpassed 2-year OS. In direct comparison, responses were deeper and more durable with the

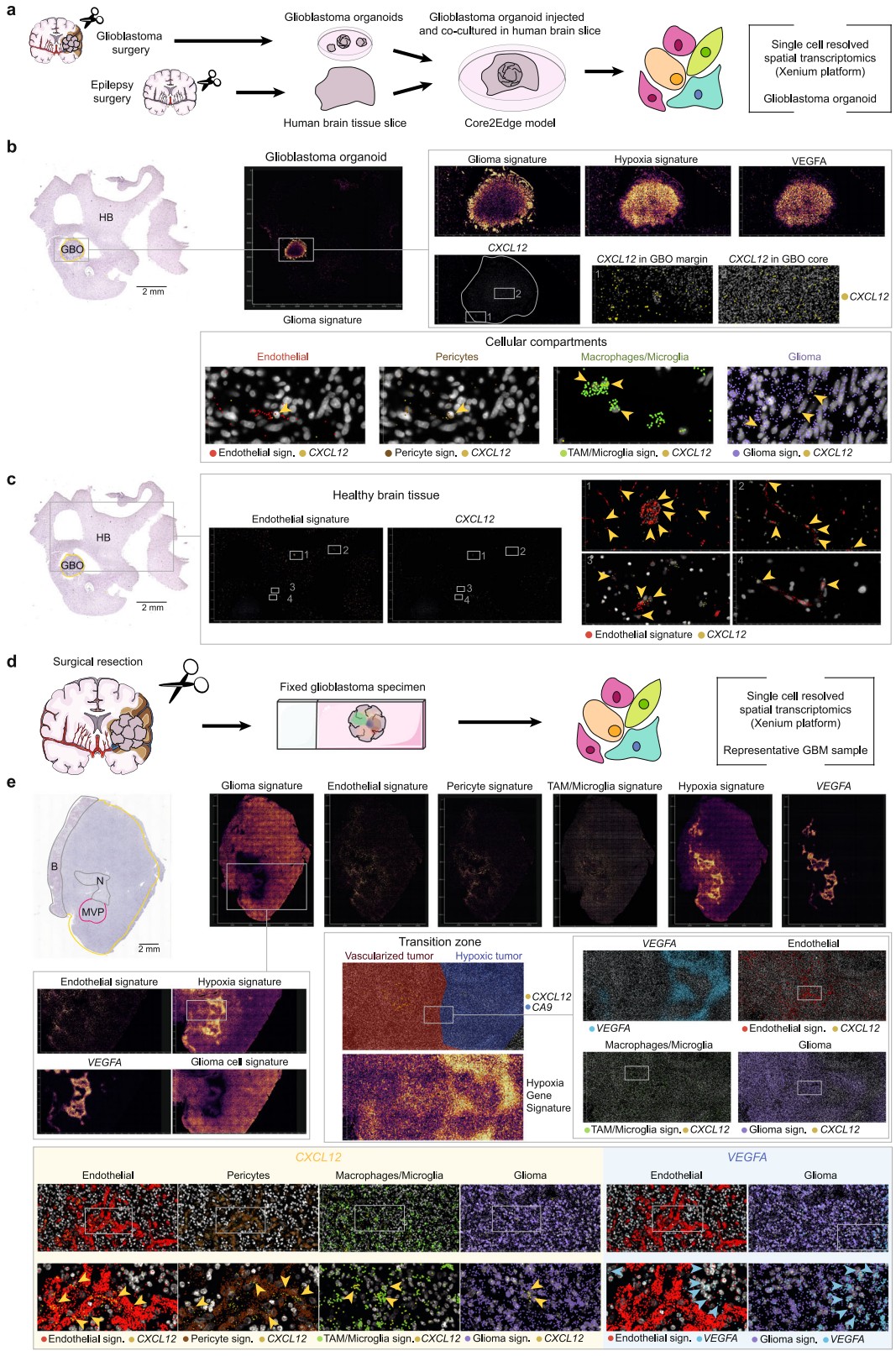

addition of BEV to NOX-A12. Under NOX-A12 only ($n = 10$), nine patients (90%) exhibited radiographic responses in terms of reduced magnetic resonance imaging (MRI) lesion sizes in at least one time point of follow-up. Of these, four patients (40%) achieved PR based on mRANO criteria ($\geq 50\%$ reduction in sum of the products of the longest perpendicular diameters, SPD).

Under NOX-A12 + BEV, all six patients achieved at least radiographic PR (Fig. 6b). In a post hoc analysis, the median best change from baseline of the highly-perfused GBM fraction (FTB^high) was − 98.2 (− 61.3 to − 100) % under NOX-A12 + BEV compared to only one patient achieving > 50% reduction in relative cerebral blood volume (rCBV) under NOX-A12 alone (Fig. 6c). Additional treatment with BEV resulted

**Fig. 3 | Spatial profiling confirms *VEGF*-hypoxia coupling and compartmentalized *CXCL12* expression in patient-derived glioblastoma organoid brain slices and freshly resected glioblastoma specimen. a** Schematic overview of the analytical workflow. Patient-derived glioblastoma cells were prepared to generate glioblastoma organoids, implanted into freshly resected human brain tissue slices and maintained in co-culture. Following fixation and paraffin embedding, samples were analyzed using single-cell-resolved spatial transcriptomics (*n* = 2 samples). **b** Upper panels: Annotated H&E staining and corresponding spatial transcriptomic signature maps of the same representative organoid-containing tissue section at increasing magnifications. Heatmaps indicate the spatial distribution of the indicated gene signatures or individual genes. Dot maps depict local *CXCL12* expression within GBO margin and core regions. Lower panels: Dot maps depicting *CXCL12*

expression across the indicated cellular compartments. Gene lists for each signature are provided in Methods. **c** Annotated H&E and spatial expression maps of the same slide as in (**b**), focusing on embedding healthy brain tissue. Insets highlight colocalization of endothelial cell signatures and *CXCL12* in healthy brain. **d** Graphical workflow for spatial transcriptomic profiling of a resected glioblastoma specimen (*n* = 1). **e** H&E staining and spatial transcriptomics signature maps of the same glioblastoma section. Upper and left panels: Heatmaps of indicated gene signatures or genes across different magnifications. Lower and right panels: Expression patterns in the transition zone with annotated regions; colored dots indicate localized gene or signature expression. B: bleeding; GBM: glioblastoma; GBO: glioblastoma organoid; HB: healthy brain; H&E: hematoxylin and eosin; MVP: microvascular proliferation; N: necrosis; sign.: signature.

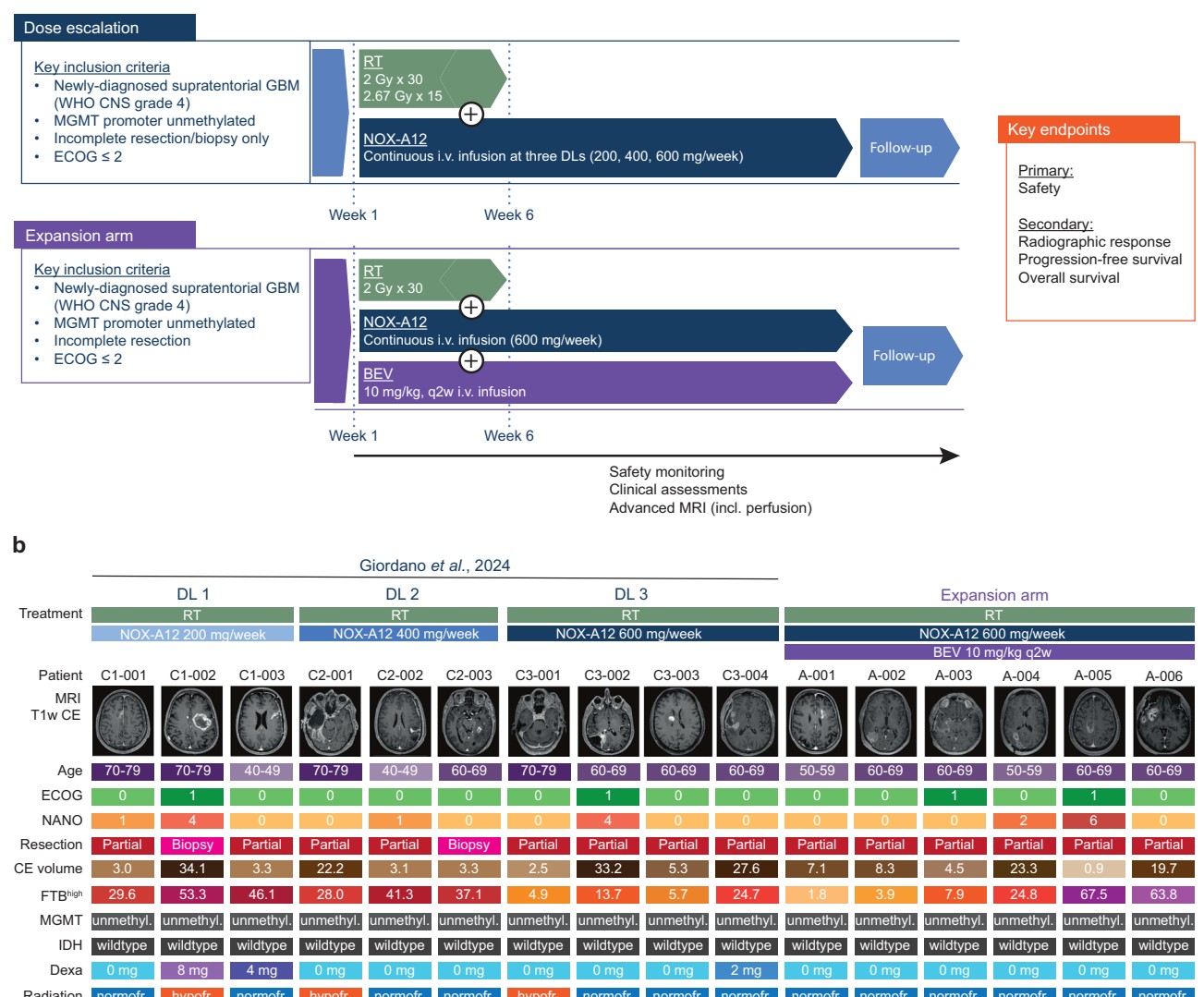

**Fig. 4 | Trial overview and individual patient characteristics. a** Graphical overview of the study. GLORIA is a multicentric phase 1/2 trial conducted to assess the safety and efficacy of RT combined with continuous i.v. treatment with NOX-A12 in newly-diagnosed, incompletely resected or biopsied GBM (CNS WHO grade 4) lacking *MGMT* promoter methylation. The initial dose escalation arm consisted of 3 escalating DLs of NOX-A12 (Giordano et al., 2024). In the expansion arm, BEV was added to the highest DL. A complete and more detailed list of eligibility criteria and outcome measures is provided at ClinicalTrials.gov, Identifier: NCT04121455. End of treatment: 26 weeks as per protocol; treatment continuation beyond 26 weeks per investigator's choice if the patient has clear clinical benefit. **b** Graphical

illustration of baseline characteristics of all GLORIA patients (*n* = 16). Age in years; CE volume in cm³. BEV: bevacizumab; CE: contrast enhancing; Dexa: daily dexamethasone dose at initiation of treatment with NOX-A12; DL: dose level; ECOG: Eastern Cooperative Oncology Group performance score; FTB^high: highly perfused fractional tumor burden; GBM: glioblastoma; IDH: IDH mutation status; *MGMT*: O⁶-methylguanine DNA methyltransferase promoter; MRI T1w CE: magnetic resonance imaging T1-weighted contrast-enhanced sequence showing residual tumor volume at baseline; NANO: neurologic assessment in neuro-oncology; NOX-A12: olaptesed pegol; RT: radiotherapy.

a

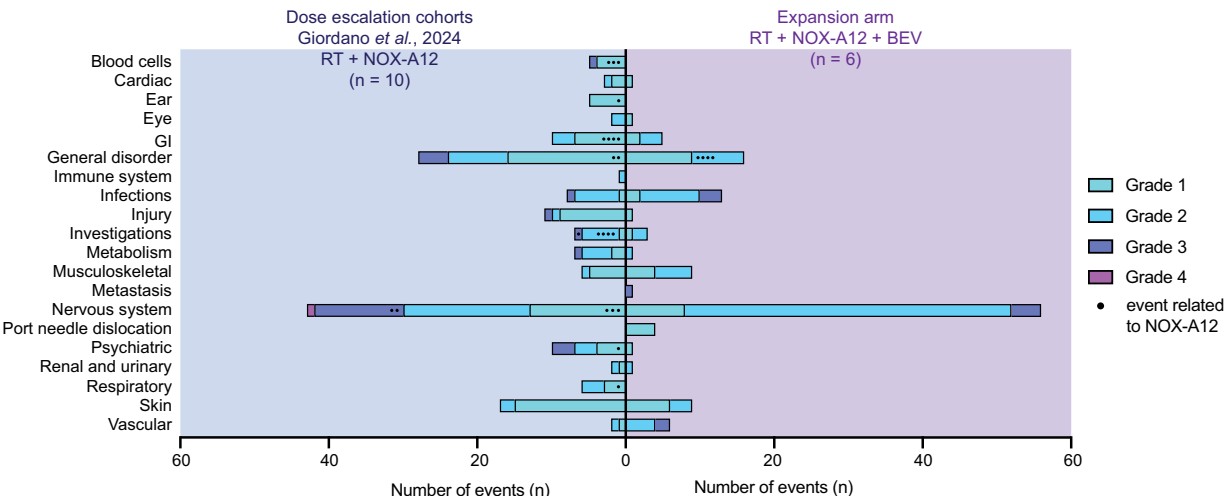

b

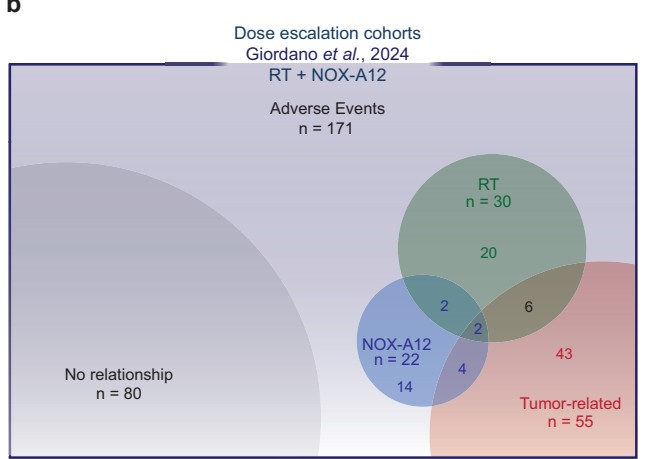
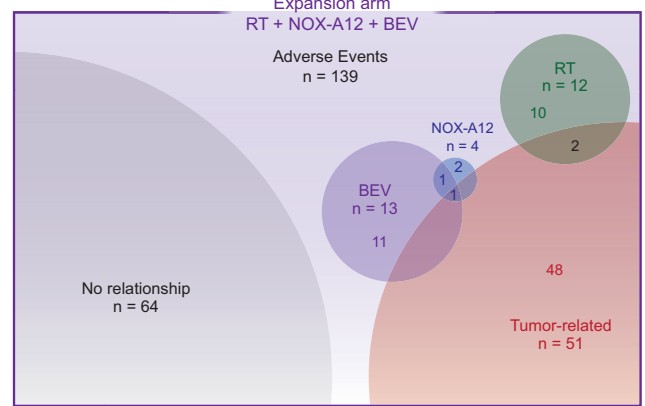

**Fig. 5 | Adverse events. a** Bar diagram illustrating the number of events and color-coded grading as per CTCAE for all adverse events grouped by system organ class. RT + NOX-A12 (*n* = 10) from Giordano et al., 2024, in blue on the left, RT + NOX-A12 + BEV (*n* = 6) in purple on the right. Events related to NOX-A12 are indicated by a black dot per event. **b** Bubble plot illustrating number and relationship of adverse events for RT + NOX-A12 (left, Giordano et al., 2024; *n* = 10) and RT + NOX-A12 + BEV (right; *n* = 6). Numbers of events with more than one underlying relationship are also indicated. Source data are provided as a Source Data file. BEV: bevacizumab; CTCAE: Common Terminology Criteria for Adverse Events; DL: dose level; GI: gastrointestinal disorders; NOX-A12: olaptesed pegol; RT: radiotherapy.

in significantly deeper tumor ($p = 0.003$) and perfusion responses ($p < 0.001$). Although apparent diffusion coefficient (ADC) responses were comparable, advanced imaging of FTB[high] again highlighted pronounced effects under NOX-A12 + BEV (Supplementary Fig. 7). Figure 6d depicts an exemplary response pattern of a patient.

## Clinical outcomes

While a PFS event was definable for 15 of the GLORIA patients, one patient needed to be censored for PFS according to the pre-defined statistical analysis plan (see patient narratives in Suppl. Note 1). Exploratory comparative analyses were performed to compare clinical outcomes between the initial dose-escalation cohorts and the expansion arm receiving additional BEV. With a median PFS of 5.7 (range 1.9–8.5, 95% confidence interval (CI) 1.9–6.0) months vs. 9.1 (range 3.9–17.0, 95% CI 7.8-not estimable) months, patients receiving triple treatment with RT + NOX-A12 + BEV showed a significantly longer PFS than patients receiving RT + NOX-A12 only (HR 0.14 (95% CI 0.02-0.60); log-rank test, $p = 0.009$; Fig. 7a). Compared to patients receiving SOC,

PFS with NOX-A12 + BEV was also significantly superior to an internal histology-matched SOC cohort (Supplementary Data 2) with a median PFS of 4.0 months (HR 0.11 (95% CI 0.02-0.41); log-rank test, $p = 0.001$). Compared to an external, contemporary multicentric SOC cohort derived from Drexler et al.[44] (Supplementary Data 3) with patient characteristics matching the GLORIA inclusion criteria, PFS with NOX-A12 + BEV was longer, but did not reach statistical significance (median PFS 6.0 months; HR 0.48 (95%CI 0.16-1.15); log-rank test, $p = 0.104$).

Importantly, patients with additional BEV treatment had a significantly longer OS (HR 0.23 (95% CI 0.05-0.79); log-rank test, $p = 0.021$; Fig. 7b) with a median OS of 12.7 (range 4.7–18.4, 95% CI 4.7-15.8) months for RT + NOX-A12 and 19.9 (range 4.1– 28.1, 95% CI 4.1-not estimable) months for the triple treatment. OS of patients receiving RT + NOX-A12 + BEV also significantly outperformed patients undergoing SOC treatment in both the internal histology-matched cohort (median OS 9.5 months; HR 0.17 (95% CI 0.04-0.53); log-rank test, $p = 0.003$) and the external SOC cohort (median OS 11.0 months; HR 0.32 (95% CI 0.09-0.88); log-rank test, $p = 0.039$).

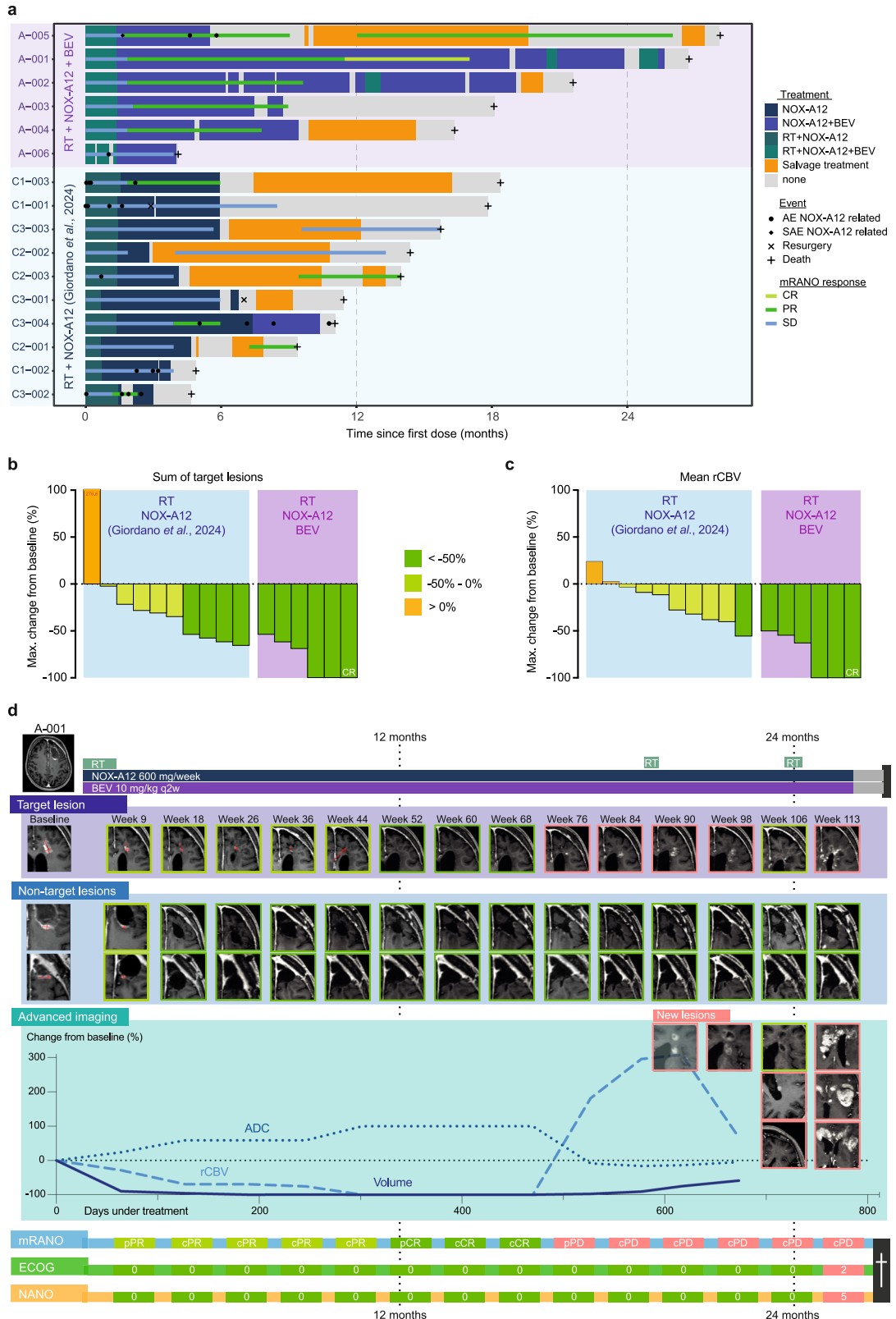

To enhance comparability with the SOC cohorts and existing literature, we additionally calculated GLORIA survival outcomes from the date of initial surgery. Both PFS and OS of patients receiving NOX-A12 + BEV were approximately 1.1 months longer, without affecting the statistical significance of comparisons with the SOC cohorts (Supplementary Fig. 8).

Given the frequent inclusion of patients with gross total resection and incomplete molecular pathology assessment in contemporary trials and real-world cohorts, the PFS and OS of GLORIA participants receiving triple therapy surpass previously reported outcomes in these populations[5,20,44–46] in the most unfavorable patient subgroup of unmethylated and incompletely resected GBM (Supplementary Fig. 9).

**Fig. 6 | Clinical response. a** Swimmer plot showing individual clinical courses of GLORIA patients ($n=16$) from the first dose (months). Patients receiving RT + NOX-A12 + BEV ($n=6$) shown in purple and RT + NOX-A12 ($n=10$, Giordano et al., 2024) in blue. Bar colors indicate ongoing treatment. Overlay lines indicate mRANO response status ranging from stable disease (SD) to complete remission (CR). Symbols indicate clinical events as specified in the legend. **b, c** Waterfall plots showing best radiographic response under experimental treatment (maximum percentage change from baseline) based on the sum of perpendicular diameters (SPD) of target lesions on contrast-enhanced T1-weighted MRI (**b**) and mean relative cerebral blood volume (rCBV) (**c**). Color gradients indicate response magnitude. Patients are grouped into RT + NOX-A12 (blue, $n=10$) and RT + NOX-A12 + BEV (purple, $n=6$). Two patients without a measurable target lesion were evaluated using the non-target-lesion response. **d** Representative treatment course of a patient achieving CR. The patient received RT (60 Gy in 2 Gy fractions), continuous NOX-A12 infusion and biweekly BEV, reaching partial remission at week 9. Treatment continued beyond week 26 due to sustained clinical benefit, with CR

achieved at week 52. Radiographic progression occurred at week 76 (PFS 17.0 months); treatment was continued due to clinical benefit and supplemented by focal salvage RT. Treatment was discontinued following clinical deterioration at week 112; overall survival was 26.8 months. The top panel illustrates the treatment timeline (gray indicates no treatment). Middle panels show longitudinal MRI of measurable lesions with percentage change from baseline for apparent diffusion coefficient (ADC; dotted line), rCBV (dashed line), and tumor volume (solid line). Bottom panel shows longitudinal mRANO, Eastern Cooperative Oncology Group performance score (ECOG), and neurologic assessment in neuro-oncology (NANO) assessments. Source data are provided as a Source Data file. AE: adverse event; BEV: bevacizumab; CR: complete remission; DL: dose level; mRANO: modified Response Assessment in Neuro-Oncology; MRI: magnetic resonance imaging; olaptesed pegol; OS: overall survival; PFS: progression-free survival; PR: partial response; rCBV: relative cerebral blood volume; RT: radiotherapy; SAE: serious adverse event; SD: stable disease; SPD: sum of perpendicular diameters.

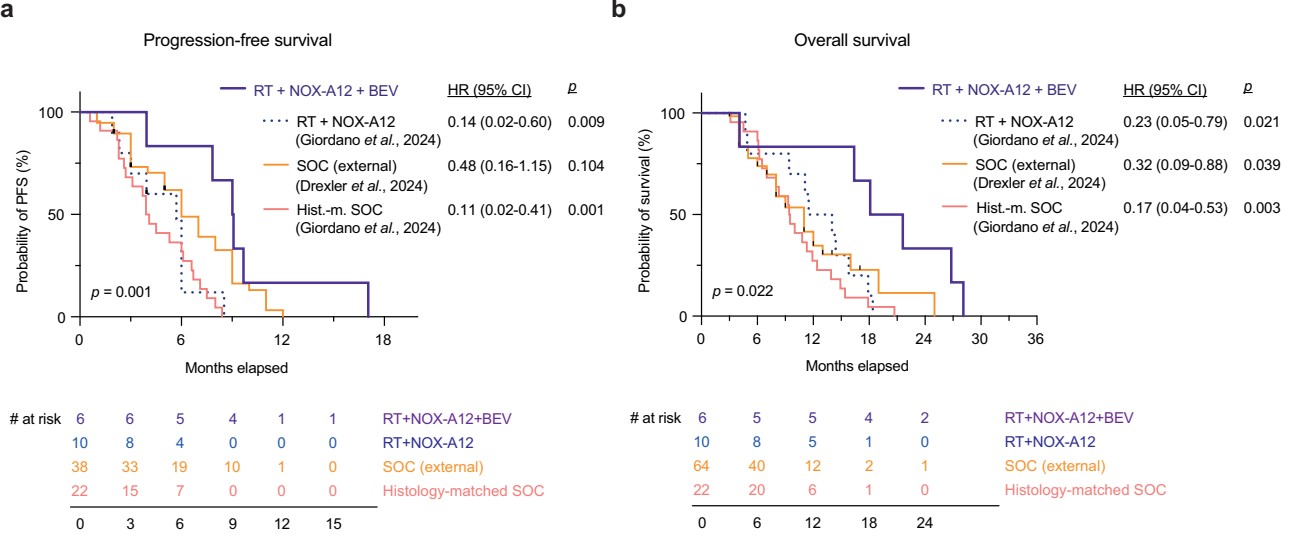

**Fig. 7 | Oncologic outcomes.** Kaplan-Meier curves of progression-free (**a**) and overall survival (**b**) in months in the GLORIA trial for patients receiving RT + NOX-A12 (dotted blue line; $n=10$; Giordano et al., 2024) versus RT + NOX-A12 + BEV (continuous purple line; $n=6$). Log-rank test (two-tailed) and Cox proportional hazards regression; $p$-values and hazard ratios with 95% confidence intervals are depicted in the corresponding graphs. Source data are provided as a Source Data File. BEV: bevacizumab; CI: confidence interval; HR: hazard ratio; PFS: progression-free survival; RT: radiotherapy.

## Correlation of outcomes with CXCL12 expression

We next assessed in an exploratory analysis whether outcomes of patients receiving RT + NOX-A12 + BEV correlated with CXCL12 expression. A higher frequency of CXCL12$^+$ endothelial and glioma cells (EG12) may serve as a predictive marker for CXCL12-targeted therapy, as it was significantly associated with prolonged PFS under NOX-A12, but not in a SOC cohort[38]. In the dose-escalation part of the trial with patients receiving RT + NOX-A12, EG12 correlated significantly with PFS ($r_s=0.865$, $p=0.005$). This correlation was lost in patients receiving RT + NOX-A12 + BEV (Suppl. Fig. 10), with a negative trend between EG12 and both PFS ($r_s=-0.714$, $p=0.136$) and OS ($r_s=-0.371$, $p=0.497$).

## Discussion

We demonstrate here that inhibition of two complementary, non-redundant and spatially distinct revascularization mechanisms, angiogenesis (VEGF-driven) and vasculogenesis (CXCL12-driven), during and after radiotherapy is a safe and promising therapeutic strategy in chemotherapy-resistant GBM.

Patients receiving dual inhibition by NOX-A12 + BEV experienced longer and deeper tumor responses as well as improved overall clinical status compared to those treated with NOX-A12 only. While

the PFS observed in GLORIA patients receiving BEV is consistent with outcomes reported in recent (larger-scaled) clinical trials of the VEGFA-targeting drug[20–22], the combinatory approach resulted in a median OS of 19.9 months from NOX-A12 initiation and 2/6 patients surpassing 2 year-OS, suggesting a trend toward improved long-term outcomes. Direct comparisons remain challenging, as the referenced trials enrolled overall more prognostically favorable patient populations. We consider the here reported GLORIA results a robust OS signal, especially since we only included GBM patients with highly unfavorable prognosis. Both PFS and OS in GLORIA patients receiving NOX-A12 + BEV appear also favorable to comparable contemporary external SOC cohorts. However, the high frequency of PFS censoring in these[44] may have introduced bias, potentially leading to an overestimation of the true PFS, thereby disadvantaging the GLORIA cohort in direct comparison. Notably, the median time from surgery to treatment initiation -marking day 1 in OS determination- was relatively long in GLORIA (1.1 months). In contrast, many studies initiate treatment earlier or commonly define the date of surgery as the starting point for outcome assessment. Consequently, the *per-protocol* GLORIA survival data should be interpreted with caution, as it likely underestimates survival compared with most contemporary trials.

We also detected physiological CXCL12 expression in healthy human brain vessels, in line with an emerging role of the CXCL12-CXCR4 axis for early-response to cerebral ischemia in conditions like cerebral insults, as shown in mouse models[47,48]. It appears likely that comparable damage and repair mechanisms are activated in GBM, where angiogenesis and vasculogenesis appear to be simultaneous, yet spatially distinct and likely complementary mechanisms. This is supported by our findings in publicly available GBM datasets and multi-omics analyses of representative GBM patients, as well as by the translational assessment of tissue from GLORIA patients. The frequency of CXCL12 expression in endothelial and glioma cells was associated with therapeutic benefit exclusively in patients receiving CXCL12-targeted therapy only. This association was no longer observed in patients who also received VEGF-targeted treatment, supporting the concept of cooperative, but functionally and spatially distinct signaling between the CXCL12 and VEGF pathways, suggesting that potential predictive biomarkers for CXCL12 inhibition may not be applicable to a combination therapy.

Both phase 3 trials investigating BEV as a first-line treatment for GBM demonstrated improved PFS but failed to demonstrate an OS benefit, raising the possibility of an unidentified treatment-induced escape mechanism. Myeloid cells have been long hypothesized to play a pivotal role in this process[25]. This hypothesis is strongly supported by findings from the North American Brain Tumor Consortium trial[49], which showed that VEGF inhibition by aflibercept induced recruitment of tumor-infiltrating myeloid cells and a subsequent release of MMP9. MMP9, in turn, facilitated a pro-invasive and pro-angiogenic escape from anti-VEGF therapy. Preclinical studies in orthotopic GBM models further demonstrated enhanced efficacy of BEV when combined with CXCL12 inhibition by NOX-A12[50]. Our findings support this hypothesis, suggesting that CXCL12 contributes substantially to pro-tumorigenic remodeling of the TME[51] and, ultimately, GBM recurrence after RT[35,52,53]. Consistent with this, increased CXCL12 and CXCR4 expression has been observed in tumor tissue of patients with recurrent GBM following BEV treatment[54]. This suggests that a CXCL12-mediated vasculogenic switch may serve as a key tumor escape mechanism to VEGF-directed therapies.

As it is inherent to the design of early-phase clinical trials, the GLORIA trial was conducted with a limited cohort size to ensure feasibility while prioritizing the assessment of safety, tolerability, and preliminary efficacy. This deliberate restriction in patient numbers aligns with the exploratory objectives of phase 1/2 trials, where careful monitoring of novel therapeutic agents in highly controlled settings is critical. While the small sample size limits the statistical interpretability and power to evaluate the survival impact of compartmental CXCL12 expression within this subset, the results still provide important hypothesis-generating insights. Our results will thus require further confirmation in larger cohorts with accompanying translational analyses to assess individual TME-based response markers. In a future phase 2/3 trial, particular emphasis should be placed on the inclusion of adequate control arms and rigorous radiographic response assessment. Given the vascular mechanism of the therapeutic approach presented here, advanced MRI techniques will be essential to reliably distinguish true response from true progression.

Ethical considerations also guided the inclusion criteria for this trial, which exclusively enrolled patients with incompletely resected, MGMT-unmethylated GBM. These patients represent a population with a dismal prognosis, for whom the current SOC with TMZ offers no established survival benefit[2,4]. This selection allowed us to address an urgent unmet need while minimizing risks to patients who might otherwise benefit from SOC. There is, however, no mechanistic reason to question the mode of action of NOX-A12 (and anti-VEGF) in MGMT-methylated or completely resected GBM. Thus, confirmation trials are now warranted that will continue to assess patient outcome independent of clinical and histopathological GBM features, but directly comparing the effects of NOX-A12, its combination with BEV and SOC in a randomized manner.

In conclusion, our findings identify VEGF and CXCL12 as complementary yet spatially distinct drivers of GBM revascularization, each contributing non-redundantly to tumor recurrence. These mechanisms underscore the complexity of the GBM tumor microenvironment and the need for multi-faceted therapeutic approaches. The clinical data from the GLORIA trial provide compelling evidence supporting the integration of RT with dual inhibition of angiogenesis and vasculogenesis as a promising strategy for the first-line treatment of GBM.

## Methods
### Trial design and oversight
GLORIA (SNOXA12C401, 2024-510964-21, NCT04121455) is a multicentric phase 1/2 study of RT in combination with NOX-A12 in first-line partially resected or unresected GBM (CNS WHO grade 4) patients with unmethylated MGMT promoter. The trial consisted of an initial dose-escalation arm and an additional expansion arm that evaluates NOX-A12 in combination with BEV. GLORIA was conducted at six academic centers in Germany, whereas the protocol was approved by the ethics committees at each participating site (ethics committees of the university hospitals of Mannheim, Bonn, Leipzig, Essen, Tübingen, and Münster). The study design and conduct complied with all relevant regulations regarding the use of human study participants. The trial followed the guidelines of the Declaration of Helsinki and the International Conference on Harmonization Good Clinical Practices Guidelines. Each patient provided written informed consent in accordance with established guidelines. The trial was reviewed by an independent data safety and monitoring committee. No trial participant received financial compensation. The first patient was enrolled on October 07, 2019. The last patient of the dose-escalation arm was enrolled on September 2, 2021. The last patient of the expansion arm with BEV was enrolled on May 23, 2022. The trial was first registered on EudraCT upon approval by the German authority (Bundesinstitut für Arzneimittel und Medizinprodukte (BfArM)) in May 2019 and transitioned to CTIS on August 2024. The dose escalation was designed as a modified 3 + 3 rule-based design according to Le Tourneau et al.[55] with three successional cohorts consisting of three patients each. Patients of DL 1 were to be treated with a weekly dose of 200 mg, DL 2 with a weekly dose of 400 mg, and DL 3 with a weekly dose of 600 mg NOX-A12. After four weeks of treatment of the first patient of DL 1, the data safety monitoring board (DSMB) reviewed all DLTs, AEs, and relevant laboratory values. During the following ten weeks of treatment, the DSMB was kept continuously informed about all DLTs and SAEs, and, at the end of this period, reviewed all DLTs, AEs, and relevant laboratory values, including NOX-A12 plasma concentrations prior to enrollment of the next two patients of this DL. The evaluation was repeated prior to enrolling patients in DL 2 and after patients 2 and 3 received at least four weeks of treatment. The same procedures were performed prior to the enrollment of further patients in DL 2 and DL 3. If none of the three patients in any DL experienced a DLT, another three patients were to be treated at the next higher DL. However, if one of the three patients in a DL experienced a DLT, three more patients were to be treated at the same DL. The dose escalation was planned to be continued until at least two patients among a cohort of three to six patients experienced DLT (i.e., ≥ 33% of patients with a DLT at that DL), but the dose would not be escalated beyond 600 mg/week. The recommended phase 2 dose (RP2D) was defined as the DL just below this toxic DL, or 600 mg/week, if this DL is not toxic. Six patients were to be included into the expansion arm. After 4 weeks of treatment of the first patient with NOX-A12 in combination with BEV, the DSMB reviewed all DLTs, all AEs, and relevant laboratory values. DLTs, according to the common terminology criteria for adverse events (CTCAE, version 5.0), were defined as any grade 3–4 non-

hematological toxicities (excluding grade 3 vomiting and/or nausea, if encountered without adequate and optimal prophylactic therapy), at any DL, assessed by the investigator and/or the sponsor as related to NOX-A12.

Inclusion criteria of the dose-escalation arm of the trial were age ≥18 years, incompletely resected or biopsied GBM (detectable post-operative residual tumor), absence of MGMT promoter (hyper) methylation, Eastern Cooperative Oncology Group (ECOG) performance score ≤2, estimated life expectancy ≥3 months, stable or decreasing dose of corticosteroids and adequate hepatic and renal function. Inclusion criteria of the expansion arm with BEV (Arm A) additionally allowed the inclusion of patients with fully resected tumors. The full listing of inclusion and exclusion criteria are provided in Supplementary Note 2. Sex was determined based on self-report. However, no sex-specific analyses were performed because the study was not powered to detect sex-related differences, and no a priori hypotheses regarding sex effects were defined. All patients were neuropathologically confirmed as GBM, IDH-wildtype (CNS WHO grade 4) according to the 2021 WHO classification for CNS tumors. If patients were ≤54 years of age, IDH mutations were additionally assessed by pyrosequencing of the IDH1 and IDH2 loci. A minimally redacted version of the study protocol is provided in Supplementary Note 2.

## Treatment and endpoints

Following adequate cranial wound healing and implantation of a venous port catheter, treatment with NOX-A12 was initiated within six weeks post cranial surgery. In the dose escalation cohorts, after an initial dose of 70, 160 or 230 mg per day, respectively on day 1, patients were administered a fixed dose of 200, 400 or 600 mg NOX-A12 per week (DL 1, DL 2, DL 3) by continuous (24 h) i.v. infusion over a commercially-available closed pump system (CADD®-Solis VIP Ambulatory Infusion Pump by Smiths Medical) starting on day 1. Treatment with NOX-A12 ended after 26 weeks. In Expansion Arm A, patients received BEV by i.v. infusion at doses of 10 mg/kg every 2 weeks for 26 weeks. Patients with disease progression during the 26-week treatment period continued treatment with all assessments if deemed appropriate by the investigator. Continuation of treatment with NOX-A12 and, if applicable, also BEV beyond 26 weeks was allowed as per each investigator's decision, if the patient had a clear clinical benefit. No simultaneous systemic oncologic treatment was permitted. Baseline patient and treatment characteristics are enlisted in Supplementary Data 1. Clinical and radiographic follow-up assessments included standard and advanced MRI sequences. The primary endpoint of the trial was safety as per the incidence of AEs. Secondary endpoints included NOX-A12 plasma levels, MTD, RP2D, imaging parameters with a specific emphasis on monitoring re-vascularization, topography of recurrence, PFS, OS and clinician/patient reported outcomes (CRO/PRO). Post hoc comparative analyses were performed to evaluate radiographic and clinical outcomes, as well as baseline CXCL12 expression profiles, between the initial dose-escalation cohorts and the expansion arm receiving additional BEV. Topography of recurrence as well as clinician- and patient-reported outcomes (NANO, QoL) were collected but are not included in the present analysis and will be reported separately. RT was initiated on day 2 after the start of NOX-A12 (and BEV) and administered as intensity-modulated, image-guided RT. Patients in the dose escalation arm received RT in a normo-fractionated (2 Gy per fraction) or hypofractionated (2.67 Gy per fraction) fashion up to cumulative doses of 60 or 40.05 Gy, respectively. For the expansion arm, all patients underwent normo-fractionated RT with 2 Gy per fraction to a cumulative dose of 60 Gy. For treatment planning, pre- and post-surgery MRI scans were co-registered on planning computer tomography (CT) scans. Gross tumor volumes (GTV), clinical target volumes (CTV) and planning target volumes (PTV) were defined as per current guidelines[56].

## Assessment of clinical and radiographic response

Patients visited the study site once weekly when presenting for the change of the medication cassette of the pump. Clinical routine follow-up visits included regular AE monitoring, physical and neurological assessments, vital signs, ECG and blood tests. AEs were assessed and graded by the investigators according to the National Cancer Institute CTCAE, version 5.0. Baseline MRIs were obtained within a week prior to treatment initiation and up to six weeks post-surgery or post-biopsy. Follow-up MRIs were obtained every eight weeks under treatment and at EOT. MRI imaging sequences included: 3D T1-weighted volumetric imaging (3D T1), T2-fluid-attenuated inversion recovery (FLAIR) imaging, diffusion-weighted imaging (DWI), T1-weighted dynamic contrast-enhanced perfusion imaging (DCE), T2-weighted turbo spin-echo imaging (T2 TSE), T2-weighted dynamic susceptibility contrast-enhanced perfusion imaging (DSC), and post-contrast 3D T1 imaging. The following additional advanced imaging parameters were calculated: DWI-derived ADC; diffusion susceptibility contrast (DSC)-derived leakage-corrected normalized rCBV and threshold-calculated FTB$^{high}$ (rCBV > 1.75); dynamic contrast-enhanced (DCE)-derived transfer constant of contrast agent (Ktrans) between the blood and the extravascular extracellular space (EES), fractional EES volume ($v_e$), and fractional plasma volume ($v_p$). All image post-processing and interpretation were performed using IB Neuro™ (Imaging Biometrics), Olea Sphere (Olea Medical), and Mint Lesion™ (Mint Medical GmbH) software and assessed by a central reader not involved in the treatment of the patients (S.B.). MRI response values for all patients can be found in the Source Data File.

## Outcome assessment

All MRI images were uploaded to an imaging database and the outcome was centrally assessed by a board-certified radiologist with expertise in the field, blinded for study site and clinical status. Target lesions (TLs) and non-target lesions (NTLs) were identified, validated and assessed in regard to tumor size (SPD) and corresponding time-point tumor response according to the modified Response Assessment in Neuro-Oncology (mRANO)[57]. Two patients enrolled had a singular residual tumor lesion meeting the inclusion criteria, while not qualifying for a target lesion (<10 mm in at least one diameter as per mRANO), thus documented as NTL. New non-measurable contrast-enhancing lesions only constituted progression in the case of a complete response (CR). NTLs only impacted the response assessment in the case of a complete response of TLs. Preliminary tumor progression (PD) or regression (PR) required confirmation in a successive scan after eight weeks. PFS was calculated as the time (in days) between the first day of treatment with NOX-A12 to the day of PD. The PFS event was defined as the first date at which progression criteria had been met, i.e., (1) the date of the first sequentially confirmed MRI assessment resulting in preliminary PD or (2) the date of the radiographic assessment irrespective of its outcome in case of simultaneous investigator-assessed clinical progression attributable to no other cause apart from the tumor or; (3) the date of death by any cause if the patient died before clinical or radiographic progression. If preliminary PD was not confirmed in a sequential MRI and there was no subsequent SD, PR, or CR, the date of preliminary PD was still considered as an event for PFS if (1) the patient stopped protocol treatment due to clinical progression; (2) no further response assessments were done; or (3) the patient died due to any cause. For patients without a clinical or confirmed radiographic progression prior to a change of systemic therapy, PFS was censored at the date of initiation of a new anticancer treatment. Independent of MRI or clinical assessment, the diagnosis of PD was not established in the case of pathologically confirmed pseudo-progression after re-surgery, which was the case in one patient, where treatment with NOX-A12 was continued afterwards. OS was calculated as the time from the first day of treatment with NOX-A12 until death by any cause. All survival data were calculated in months,

rounded to one decimal place. The individual clinical courses, therapies, investigator decisions, and definitions of PFS and OS events for all patients are described in detailed narratives provided in Supplementary Note 1.

### Internal and external standard-of-care (SOC) cohorts

We benchmarked oncological outcomes of the GLORIA trial to an internal (University Hospital Bonn)[38] and a recently published external German multicenter cohort[44] of GBM patients with comparable patient characteristics resembling the inclusion criteria of the GLORIA trial treated with standard RT and optional TMZ. The procedures were approved by the Ethics Committee of the University Hospital Bonn (approval number: 222/23-EP). The local reference cohort includes patients treated at the University Hospital Bonn between 2010 and 2023. Histological criteria for selection of reference patients were: newly-diagnosed GBM, IDH-wildtype (CNS WHO grade 4) according to the valid WHO classification for CNS tumors, absence of *MGMT* promoter methylation as confirmed by pyrosequencing[58]. Clinical criteria were: ECOG of 0-2, status post biopsy or incomplete resection, and first-line therapy with RT (and optionally TMZ). Data to establish the external German multicenter SOC cohort was extracted from publicly available annotated data[44], which provided data of IDH-wild type glioblastoma patients undergoing surgical resection and sequential chemoradiotherapy between 2016 and 2023 from three German neuro-oncological centers (Hamburg, Frankfurt, Berlin). The patients selected for the SOC cohort had the following characteristics closely resembling the GLORIA trial: IDH-wild type glioblastoma, ECOG 0-2, incomplete resection (<90% removal), unmethylated *MGMT* promoter. The patient characteristics of both cohorts are provided in Supplementary Data 2 and 3 and the Source Data File.

### Plasma NOX-A12 and CXCL12 concentrations

A Liquid Chromatography-UV assay, based on an anion-exchange chromatography analysis coupled to a UV detector, was used to detect and quantify the analyte NOX-A12 in patient plasma samples. The comparable calibration curve, generated from a standard of dilution series, corresponded to a linear range of NOX-A12 concentrations in human plasma from 0.5 to 200 µg/mL (0.034–13.6 µM). Quantification of CXCL12 concentrations in patient plasma samples was performed by a contractor (Swiss BioQuant AG) using HPLC-MS/MS bioanalytics. The comparable calibration curve corresponded to a linear range of CXCL12 (human CXCL12α and CXCL12β) concentrations in human plasma from 25 to 2500 nM.

### Multiplexed immunofluorescence of tumor tissue

To assess cellular distribution patterns of hypoxia and CXCL12, human GBM samples were assessed by multiplexed immunofluorescence. Histological criteria for selection of these samples were: newly-diagnosed GBM, IDH-wildtype (CNS WHO grade 4) according to the valid WHO classification for CNS tumors, absence of *MGMT* promoter methylation as confirmed by pyrosequencing[58]. All tumor samples were obtained following informed consent as part of SOC surgical procedures, and all patients had consented to analyses of their tissue. The scientific procedures were approved by the Ethics Committee of the University Hospital Bonn (approval number: 222/23-EP). Formalin-fixed paraffin-embedded (FFPE) tumor samples were sliced by standard procedures at 3 µm slice thickness and adhered onto poly-L-lysine-coated coverslips. Antibody conjugation, tissue staining, and mIF imaging were performed (with modifications) as described elsewhere[59,60]. In short, purified, carrier-free antibodies were conjugated to maleimide-modified oligonucleotides (Biomers), concentrated, reduced, and washed with buffer. Maleimide-modified oligonucleotides were first dissolved on 1x DPBS, then added to the reduced antibody and incubated at room temperature for two hours in a 2:1 (w/w) ratio with the antibodies. Next, the conjugated antibodies

were washed in high-salt PBS three times and then eluted by centrifugation at 3000 × g for 2 min in the CODEX® antibody stabilizer solution. The conjugated antibodies were stored at 4 °C until usage. Prepared FFPE tissues were baked at 55 °C for 30 min, and rehydrated by immersion in fresh xylene twice, for 5 min and in descending concentrations of ethanol, each step for 5 min (100% twice, 95% twice, 70%, ddH2O twice). Heat-induced epitope retrieval was performed using 1 x Dako target retrieval solution, pH 9 (Agilent) at high pressure, for 20 min. Tissues were then washed for 10 min in 1 x TBS IHC wash buffer with Tween 20 (Thermo Fisher, 28360). Tissues were blocked for 1 h at room temperature using 100 µl of blocking buffer. Conjugated antibodies were added to the blocking buffer, concentrated through a 50 kDa Amicon Ultra Filter and resolved in the blocking buffer. Tissues were incubated with the antibody staining solution in a humidity chamber overnight at 4 °C. The following antibodies were used: Ki-67, clone B56, BD Biosciences, Cat.# 556003 (RRID:AB_396287); SDF1/CXCL12, clone 79018, Thermo Fisher, Cat.# MA5-23759 (RRID:AB_2608711); α-SMA, clone 1A4, ThermoFisher, Cat.# 14-9760-82 (RRID:AB_2572996); CD31, clone EP3095, Abcam, Cat.# ab226157; GFAP, clone 2.2B10, Thermo Fisher Scientific, Cat.# 13-0300 (RRID:AB_2532994); CD68, clone KP-1, Biolegend, Cat.# 916104 (RRID:AB_2616797); CA9, polyclonal, NovusBio, Cat.# NB100-417. After staining, tissues were washed twice in S2 buffer and fixed with a three-step fixation process. First, tissues were fixed in S4 containing 1.6% paraformaldehyde (Electron Microscopy Science, 15710-S) for 10 min, followed by a 15 min long incubation in 100% ice-cold methanol (Sigma, 34860-1L-R) for 5 min, and a final fixation with BS3 fixative solution dissolved in 1x PBS at room temperature for 20 min. Tissues were stored in S4 in a 6-well plate at 4 °C for up to two weeks, or further processed for imaging. 400 nM fluorescent oligonucleotide working solution was aliquoted in CorningTM black 96-well plates (Merk, CLS3925-100EA) in 250 µl of plate buffer, according to the multi-cycle reaction panel. Image acquisition was performed on Zeiss Axio Observer 7 microscope equipped with a Colibri 7 LED Light source (Carl Zeiss), and a Prime BSI PCIe camera (Teledyne Photometrics). Imaging cycles were performed using an Akoya Phenocycler™ instrument and CODEX® instrument manager software (Akoya Biosciences). Automated images were acquired with the Plan-Apochromat 20X/0,8 M27 (a = 0.55 mm) objective (Carl Zeiss), and the imaging pipeline was controlled by a focus strategy with autofocus for each support point created, with a three z-stack image with a distance of 1.5 µm. DAPI (1:300 final concentration) was imaged in each cycle at an exposure time of 20 milliseconds and an LED intensity of 40%. The images were processed with CODEX® Processor (Akoya Biosciences, v.1.8.3.14) and analyzed with HALO® Image Analysis software (Indica Labs, v.3.3.2541.256). After each multi-cycle reaction, standard H&E staining was performed on the same tissue slice to confirm histopathological features. The H&E staining was analyzed independently by a neuropathologist with 5 years of experience and a board-certified neuropathologist with more than 30 years of experience in the field. With consensus, pathological features of GBM (CNS WHO grade 4) were again confirmed, and zonal characteristics within the tumor tissue were depicted. Adjacent regions like leptomeninges, hemorrhage or healthy brain tissue were excluded and only confirmed tumorous tissue parts were considered for the following analyses. Annotated H&E stainings can be found in Supplementary Fig. 11 and Supplementary Fig. 12. Analyses were performed using the Highplex FL module (v. 4.1.2) from HALO®. A nucleus/cytoplasm membrane % completeness threshold for positivity was set as follows: DAPI, Nucleus % Completeness Threshold 15%; CXCL12, Nucleus and Cytoplasm % Completeness Threshold 30%; CD68, Nucleus and Cytoplasm % Completeness Threshold 30%; CD31, Nucleus and Cytoplasm % Completeness Threshold 30%; Ki-67, Nucleus % Completeness Threshold 35%, α-SMA, Nucleus and Cytoplasm % Completeness Threshold 35%; GFAP, Nucleus and Cytoplasm % Completeness Threshold 25%%; CA9,

**Table. 1 | Buffers and solutions for multiplexed immunofluorescence**

| Buffer | Composition |
|---|---|
| TCEP-reducing solution | 2.5 mM TCEP Sigma, 646547 |
| EDTA pH 8.0 | 2.5 mM EDTA pH 8.0 (Invitrogen, AM9261) |
| Buffer C | 150 mM NaCl (Carl Roth, 9265.2), and 2 mM Tris stock solution (Carl Roth, AE15.3), pH 7.2, NaN3 (AppliChem, A14300,1000) |
| High-salt PBS | 900 mM NaCl in 1× DPBS (Gibco, 14190-094). |
| CODEX® antibody stabilizer solution | 0.5 M NaCl, 5 mM EDTA, and 0.02% w/v NaN3 in PBS antibody stabilizer solution (CANDOR Biosciences GmbH, 131125). |
| Staining solution 1 (S1) | 5 mM EDTA, 0.5% w/v bovine serum albumin (BSA, Carl Roth, 8076.3) and 0.02% w/v NaN3 in 1 × DPBS, stored at 4 °C. |
| Staining solution 2 (S2) | 61 mM NaH2PO4 (Sigma, S0876), 39 mM NaH2PO4 · H2O (Sigma, S9638), 250 mM NaCl in a 1:0.7 v/v solution of S1 and doubly-distilled H2O (ddH2O); final pH 6.8–7.0, stored at 4 °C |
| Staining solution 4 (S4) | 0.5 M NaCl in S1, stored at 4 °C. |
| Blocking buffer | S2 buffer containing B1 (1:20), B2 (1:20), B3 (1:20), and BC4 (1:15), stored at 4 °C. |
| Blocking reagent 1 (B1) | 1 mg/ml mouse IgG (Sigma, I5381) in S2, stored at 4 °C. |
| Blocking reagent 2 (B2) | 1 mg/ml rat IgG (Sigma, I4121) in S2, stored at 4 °C. |
| Blocking reagent 3 (B3) | Sheared salmon sperm DNA (Invitrogen, AM9680), 10 mg/ml in H2O, stored at 4 °C. |
| Blocking component 4 (BC4) | Mixture of 57 non-modified oligonucleotides (Biomers, Supplementary Data 4) at a final concentration of 0.05 mM each in TE buffer (Sigma, 93302), stored at 4 °C |
| BS3 fixative solution | 200 mg/ml BS3 (ThermoFisher, 21580) in DMSO from a freshly opened ampoule (Sigma, D2650-5 x 5ML), stored at 20 °C in 3 µl aliquots. |
| H2 buffer | 150 mM NaCl, 10 mM Tris pH 7.5, 10 mM MgCl2 • 6 H2O (Carl Roth, 2189.1), 0.1% w/v TritonTM X-100 (Sigma, X-100) and 0.02% w/v NaN3 in ddH2O. |
| Plate buffer | H2 buffer containing DAPI nuclear stain (1:300, Biolegend, 422801) and 0.5 mg/ml sheared salmon sperm DNA. Fluorescent oligonucleotide stock solution (Biomers): 100 µM Fluorescent oligonucleotide dissolved in 1 × TE buffer, stored in the dark at 4 °C. |
| Fluorescent oligonucleotide working solution | Fluorescent oligonucleotide stock solution diluted 1:10 in 1 × TE buffer, stored in the dark at 4 °C. |

Nucleus and Cytoplasm % Completeness Threshold 20%. Cellular phenotypes were defined as follows: endothelial cells (DAPI + , CD31 + ), pericytes (DAPI + , α-SMA + , CD31-), macrophages/microglia (DAPI + , CD68 + ), tumor cells (DAPI + , GFAP + , CD68-, CD31-, α-SMA-). All phenotypes were also assessed for CXCL12 and CA9 expression positivity. Nuclear segmentation was based on DAPI with a custom-trained deep learning neural network algorithm, with nuclear segmentation aggressiveness of 0.5 and nuclear size for positivity set between 12 and 1000 µm2. All buffers and solutions for multiplexed immunofluorescence are provided in Table 1 below. Details on the multi-cycle reactions and oligo sequences can be found in Supplementary Data 4.

**Generation of brains slice-implanted glioblastoma organoids (Core2Edge)**

The Core2Edge model involves the implantation of patient-derived GBOs into human cortical brain slices. The procedures were approved by the ethics committee of the University of Bonn (approval no. 162/10 and 504/20), and patients provided written informed consent. Human organotypic brain slices were generated from tissue obtained during epilepsy surgery. After trimming, tissue blocks were mounted on a vibratome and sectioned into 300 µm slices in ice-cold, carbogenated brain slice preparation medium (see Supplementary Table 4). Intact slices were transferred onto Millicell® inserts (Millipore., cat. no. PICM03050) positioned at the air–liquid interface and cultured in brain slice growth medium (see Supplementary Table 4) at 37 °C and 5% CO₂ for 24 h. For the Core2Edge model, GBOs from a stable pool were selected for implantation. GBOs were generated from patient-derived tumor tissue with confirmed glioblastoma, IDH-wildtype, WHO grade 4, according to the protocol by Jacob et al.[61] Prior to implantation, GBOs were stained with CellBrite™ Green to enable visualization during the implantation (see Supplementary Table 4). Implantation was performed under a fluorescence stereomicroscope. Therefore, a small incision was created in the white matter of each organotypic slice, and a single stained GBO was gently positioned into the tissue using a microdissector. Slices were then returned to the air–liquid interface and maintained for 10 days, with medium changes every 48 h.

**Spatial transcriptomics**

Following co-culture, Core2Edge slices were fixed in 4% buffered formalin, dehydrated through graded ethanol and xylene, and embedded in paraffin using standard procedures. In addition, FFPE tissue from a glioblastoma patient was processed according to the criteria described in the section Multiplexed immunofluorescence. FFPE blocks from both the Core2Edge samples and the clinical tumor specimen were sectioned at 5 µm and prepared for spatial transcriptomics using the Xenium In Situ platform (10x Genomics). Sections were mounted within the defined sample area and processed following the Tissue Preparation Guide (CG000578, 10x Genomics), employing the Human Brain Panel (266 genes) together with 100 custom-designed probes. After probe hybridization, ligation, rolling-circle amplification, DAPI staining, and cyclic fluorescent imaging, spatial gene expression data were acquired using the Xenium Analyzer. Spatial transcriptomics data were processed and visualized using Xenium Explorer (version 4.1.0, 10x Genomics). Cell-type and pathway signatures were defined a priori based on curated gene sets derived from established literature. For each spatial feature, log-normalized expression values of the genes within a given signature were averaged to obtain a composite signature score. Cell-type signatures were defined as follows: glioma cells: *AQP4, OLIG1, OLIG2, POSTN, SOX2*; endothelial cells: *CD34, CLDN5, PECAM1, PLVAP, VWF*; pericytes: *ABCC9, ACTA2, CSPG4, PDGFRB, RGS5, TGFBI*; macrophages: *CD14, CD163, FCGR3A, CD68*; microglia: *AIF1, CX3CR1, CXCL10, IL1B, P2RY12, TMEM119, TREM2*; hypoxia signature: *ANGPTL4, CA9, DDIT3, HIF1A, LDHA, MIF, PGK1, SLC2A1*.

To generate spatial density maps, the tissue was divided into 10 × 10 µm bins, and local signal intensity was computed using kernel-smoothed density estimation. Density values were visualized using the

Inferno color scheme (low: black/dark violet; high: bright yellow/white). Heatmaps display either individual gene expression or composite signature scores across spatial coordinates. All spatial plots, overlay visualizations, and region annotations were generated in Xenium Explorer.

## Public data RNA sequencing analysis

The bulk tumor dataset Glioblastoma Multiforme from the TCGA Research Network was downloaded from cbioportal[62] under https://www.cbioportal.org/study/summary?id=gbm_tcga_pan_can_atlas_2018. The GBM dataset with regional tissue mapping from Puchalski et al.[42] was downloaded from the IVY GAP Glioblastoma Atlas Project (https://glioblastoma.alleninstitute.org/static/home). Data was analyzed and visualized using R version 4.2.2 and GraphPad Prism 10.

## Software and statistical analysis

No formal sample size calculations were performed for this trial. The dose escalation part was designed as a modified 3 + 3 rule-based design as described above and in the study protocol provided in Supplementary Note 2. No data were excluded from the analyses. The experiments were not randomized, and thus, investigators were not blinded to treatment during experiments and outcome assessment. The independent central reader was blinded to all clinical aspects of the trial. Similarly, spatial gene expression profiling and mIF were conducted in a blinded manner, with investigators unaware of clinical trial data and patient identities.

Graphical elements were generated using R version 4.2.2, GraphPad Prism 10 (GraphPad Software) and Adobe Illustrator 2023 (Adobe Inc.). Illustrations in this manuscript were in part created using Servier Medical Art, provided by Servier, and licensed under a Creative Commons Attribution 4.0 Unported License (https://smart.servier.com). Database management (eCRF) was carried out using Viedoc version 4.66 eCRF (Viedoc Technologies) and Microsoft Excel 2019 (Microsoft Corporation). Statistical tests were performed using GraphPad Prism 10 and R version 4.2.2 as specified in the figure legends. Descriptive statistics were applied to characterize the patient collectives, treatment response, and observed toxicity. Survival rates were estimated using the Kaplan–Meier method and compared by the log-rank test. Exploratory Cox proportional hazards regression analyses were additionally performed, and the proportional hazards assumption was verified using log-log survival plots and Schoenfeld residuals. Group differences for continuous variables were evaluated using the unpaired two-tailed Mann–Whitney U test. Spearman's rank correlation was used for correlation analysis. Group comparisons of drug and target concentrations at identical time points were performed using the Kruskal–Wallis test, followed by pairwise Mann–Whitney tests. Proportional data were transformed using the arcsine square-root transformation to better meet assumptions of normality required for parametric testing. For integrated pharmacokinetic evaluation, the area under the curve (AUC) was calculated for each patient. Between-group differences in AUCs were assessed using Welch ANOVA.

## Reporting summary

Further information on research design is available in the Nature Portfolio Reporting Summary linked to this article.

## Data availability

The study protocol is made available in the Supplementary Information. Source data are provided with this paper. High-resolution images of Supplementary Fig. 2 are provided in the following repository: https://doi.org/10.5281/zenodo.18674810. High-resolution images of Supplementary Fig. 11 and Supplementary Fig. 12 are provided in the following repository: https://doi.org/10.5281/zenodo.18674810. Raw data from spatial transcriptomics are provided in the following repository: https://www.ncbi.nlm.nih.gov/geo/query/acc.cgi?acc= GSE324268. The publicly available GBM dataset with regional tissue mapping used in this study from Puchalski et al.[42] can be accessed via the IVY GAP Glioblastoma Atlas Project under https://glioblastoma.alleninstitute.org/static/home. The publicly available bulk GBM tumor dataset from the TCGA Research Network[39] can be accessed under https://www.cbioportal.org/study/summary?id=gbm_tcga_pan_can_atlas_2018. Individual participant data are not publicly available due to patient privacy regulations. De-identified data underlying the findings are available from the corresponding authors for non-commercial research purposes at all times following completion of a data access and transfer agreement and approval from the institutional ethics board. In line with German regulations for clinical research, the data will be archived and remain accessible for 30 years after completion of the study. The remaining data are available within the Article, Supplementary Information or Source Data file. Source data are provided in this paper.

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

## Acknowledgements

We wish to thank all patients and their families for their commitment to participating in this study. The clinical study was sponsored by TME Pharma AG (Berlin, Germany). TME Pharma AG also provided NOX-A12 and BEV, as well as clinical trial supply related to this work. The sponsor had no role in data analysis, interpretation of the data, or preparation of the manuscript. Parts of the translational research were supported by grants from the Deutsche Forschungsgemeinschaft (DFG, German Research Foundation; SFB1319/TP-B05 to FAG). J.P.L. is supported by the German Federal Ministry of Research, Technology and Space (BMFTR; funding code (FKZ) 01EO2107). L.L.F. is supported by the BONFOR program of the Medical Faculty of the University of Bonn, Germany (grant ID 2022–1A-09), followed by the Neuro-aCSis program, funded by the DFG (grant ID 2024-12-03). M.H. is a member of EXC2151 and supported by the DFG under Germany's Excellence Strategy—EXC2151–390873048. We thank the study site staff for their ongoing support, particularly Christiane Landwehr, Joana Kömpel, Katja Klever, Monika Brüggemann, Katja Bienentreu, Mirco Muscheid, Inga Krause, Nadja Talhi, Sabrina Agkatsev, Gina Seidel. We thank Ute Heuser-Figgemeier and Alexandra Brüggemann from the Institute of Neuropathology and Sandra Bald and Simone Glees from the Department of Dermatology at the University Hospital Bonn for their help with the GBM tissue processing. The UKB histopathology research core facility is supported by the DFG under Germany's Excellence Strategy—EXC2151–390873048. We would like to thank the Microscopy Core Facility of the Medical Faculty at the University of Bonn for providing instrumentation funded by the DFG—project number 388168919.

## Author contributions

This study and the translational analyses were conceptualized and conceived by F.A.G., J.P.L. and M.H. F.A.G., J.P.L., T.Z., E.S., N.S., K.S., C.O., S.K., P.H., M.R., M.P., O.M.G., G.T., M.G., C.S. and U.H. treated trial patients. J.P.L., O.M., T.Z., J.W., D.G.P., M.V. and U.H. provided SOC patient data. S.B. and J.P.L. assessed and reviewed the patient imaging. Material preparation and data collection were performed by J.P.L. Histological staining was assessed by L.L.F. and T.P. Multiplex-immunofluorescence was performed by R.T. B.P., A.L.P. and M.S. performed brain slice-implanted GBO generation and spatial transcriptomics. Spatial transcriptomics were analyzed by J.P.L. and S.K. RNA sequencing data was analyzed by J.P.L. Computational data analysis was performed by J.P.L. F.A.G., E.G., U.H. and M.H. provided funding and resources. The first draft of the manuscript was written by J.P.L. Tables, and figures were created by J.P.L. F.A.G., M.P., U.H. and M.H. reviewed and edited the data and manuscript. All authors commented on previous versions of the manuscript. All authors read and approved the final manuscript.

## Funding

## Competing interests

F.A.G. reports travel expenses, stocks and honoraria from TME Pharma AG related to this work; research grants and travel expenses from ELEKTA AB; grants, research grants, travel expenses and honoraria from Carl Zeiss Meditec AG, OncoMAGNETx Inc. and Varian Medical Systems Inc.; travel expenses and/or honoraria from Bristol- Myers Squibb, Cureteq AG, GUERBET S.A., Roche Pharma AG, MSD Sharp and Dohme GmbH, Siemens Healthineers AG, Varian Medical Systems Inc., and Astra- Zeneca GmbH; non-financial support from Oncare GmbH and Opasca GmbH and patent US10857388B2 together with Carl Zeiss Meditec AG; all unrelated to this work. J.P.L. reports stocks and travel expenses from TME Pharma AG related to this work; travel expenses and honoraria from Carl Zeiss Meditec AG, honoraria and clinical advisory board membership from OncoMAGNETx Inc. and Biotex Inc., stocks and honoraria from Siemens Healthineers AG, and stocks from Bayer AG and BioNTech AG, all unrelated to this work. E.S. reports travel expenses and honoraria from Carl Zeiss Meditec AG, unrelated to this work. C.O. has received travel support from Novocure and honoria by Horizon and Novocure. He has received a Clinician Scientist stipend of the University Medicine Essen Clinician Scientist Academy (UMEA) sponsored by the faculty of medicine and Deutsche Forschungsgemeinschaft (DFG) and a research grant by Servier, all unrelated to this work. N.S. received honoraria from Servier, unrelated to this work. M.P. reports research contributions with Roche Holding AG related to this work. U.H. reports honoraria from Medac, Bayer, Servier and OncoMAGNETx Inc., unrelated to this work. M.H. reports travel expenses, honoraria for webinars, and research support (consumables) from TME Pharma AG related to this work. M.H. also reports honoraria and clinical advisory board membership from OncoMAGNETx Inc., unrelated to this work. The remaining authors declare no competing interests.

## Additional information

Frank A. Giordano[1,2,3,22] ✉, Julian P. Layer [4,5,22], Roberta Turiello [5], Lea L. Friker [5,6], Oriol Mirallas[7], Barbara E. F. Pregler[8], Anna-Laura Potthoff[8], Thomas Zeyen[9], Johannes Weller[9], Elena Sperk [1,10], Katharina Sahm[11,12], Christoph Oster [13,14], Sied Kebir[13,14,15], Peter Hambsch[16], Niklas Schäfer[9], Sebastian Kadzik [5], Mirjam Renovanz[17], Torsten Pietsch[6], Sotirios Bisdas [18], Juan Manuel Sepúlveda-Sánchez[19], Diego Gómez-Puerto[7], Maria Vieito[7], Michael Platten [11,12], Eleni Gkika[4], Oliver M. Grauer [20], Ghazaleh Tabatabai [17], Matthias Schneider [8], Martin Glas[13,14,21], Clemens Seidel [16], Ulrich Herrlinger [9,22] & Michael Hölzel [5,22] ✉

[1]Department of Radiation Oncology, University Medical Center Mannheim, University of Heidelberg, Mannheim, Germany. [2]DKFZ-Hector Cancer Institute at the University Medical Center Mannheim, Mannheim, Germany. [3]Mannheim Institute for Intelligent Systems in Medicine (MIISM), University of Heidelberg, Mannheim, Germany. [4]Department of Radiation Oncology, University Hospital Bonn, University of Bonn, Bonn, Germany. [5]Institute of Experimental Oncology, University Hospital Bonn, University of Bonn, Bonn, Germany. [6]Institute of Neuropathology, University Hospital Bonn, University of Bonn, Bonn, Germany. [7]Medical Oncology Department, Vall d'Hebron Institute of Oncology, Barcelona, Spain. [8]Department of Neurosurgery, University Hospital Bonn, University of Bonn, Bonn, Germany. [9]Department of Neurooncology, University Hospital Bonn, University of Bonn, Bonn, Germany. [10]Mannheim Cancer Center, University Medical Center Mannheim, University of Heidelberg, Mannheim, Germany. [11]Department of Neurology, Medical Faculty Mannheim, University of Heidelberg, Mannheim, Germany. [12]DKTK Clinical Cooperation Unit Neuroimmunology and Brain Tumor Immunology, German Cancer Research Center, Heidelberg, Germany. [13]Department of Neurology, University Medicine Essen, University Duisburg-Essen, Essen, Germany. [14]DKFZ-Division Translational Neurooncology at the West German Cancer Center (WTZ), University Medicine Essen, University Duisburg-Essen, Essen, Germany. [15]German Cancer Research Center (DKFZ), Heidelberg, Germany. [16]Department of Radiation Oncology, University of Leipzig Medical Center, University of Leipzig, Leipzig, Germany. [17]Department of Neurology and Interdisciplinary Neuro-Oncology, University Hospital Tübingen, Eberhard Karls University, Tübingen, Germany. [18]Lysholm Department of Neuroradiology, National Hospital for Neurology and Neurosurgery, London, United Kingdom. [19]Medical Oncology Department, University Hospital 12 de Octubre, Madrid, Spain. [20]Department of Neurology, University Hospital Münster, University of Münster, Münster, Germany. [21]Department of Neurology and Neurooncology, St. Marien Hospital Lünen, Lünen, Germany. [22]These authors contributed equally: Frank A. Giordano, Julian P. Layer, Ulrich Herrlinger, Michael Hölzel. ✉e-mail: Frank.Giordano@umm.de; michael.hoelzel@ukbonn.de

