## [Transparent Peer Review file · Nature Communications]

L-RNA aptamer-based CXCL12 inhibition combined with radiotherapy and bevacizumab in newly-diagnosed glioblastoma: expansion of the phase I/II GLORIA trial

Corresponding Author: Professor Frank Giordano

Version 0:

Reviewer comments:

Reviewer #1

(Remarks to the Author)

This is a well-written follow up manuscript of the GLORIA trial reporting a 6 patient safety expansion cohort of CXCL12 inhibition combined with Bevacizumab and radiotherapy in newly diagnosed GBM.

The authors provide the rationale of the combination with spatially resolved data obtained from two publically available data sources (TCGA and the Ivy Glioblastoma Atlas Project). This suggests distinct regulatory mechanisms of vascularization in GBM which the authors then explore using multi-colour IF in 5 human GBM samples obtained via SOC surgical procedures. Multi-colour IF does confirm spatially distinct expression of CXCL12 and CA9 (surrogate of hypoxia) supporting their hypothesis.

The authors then report the clinical course of the additional 6 patients treated on the expansion cohort (the escalation data having been previously already published - Giordano 2024. Bevacizumab was given at 10mg/kg -the dose previously used in the Avaglio studies. No additional safety signals were seen, albeit numbers too small to make conclusions. The outcomes of these 6 patients due appear to be better than expected with standard of care, in particular given the poor prognosis patient population enrolled (incomplete resection, PS<2) although I note steroid doses are not reported. No translational data from these 6 patients is actually presented - have the authors got access to these samples as baseline multi-colour IF on these samples would have significantly strengthened the paper.

In conclusion, the authors provided a solid rationale for the exploration of their combination hypothesis - however I am disappointed that there were no mechanistic attempts to explore this in vitro (for example using brain slice cultures from their SOC surgical samples) which would have provided proof of concept given their spatial IF assays are already established. The preliminary clinical data from 6 patients treated in an expansion cohort is intriguing but too small to draw any firm conclusions.

Reviewer #2

(Remarks to the Author)

Overall the manuscript is very well written and quite interesting. I have the following questions:

Are there differences in the clinical and surgical traits between the external contemporary multicentric SOC cohort and the internal histology-matched SOC? Why was progression-free survival (PFS) data only available for 39 out of 64 patients in the external cohort? Could this skew the results, making the median PFS appear higher?

It appears that there is a significant confounder, as patients in Arm B of the 6-arm expansion may have undergone a gross total resection (GTR). Why was this permitted?

What was the approach to censoring patient C2-001? It appears that medians and ranges were calculated using an Excel

spreadsheet. Was this the method for conducting the survival analysis? It's unusual to report ranges for median survival; generally, the median from a Kaplan-Meier estimate should include the corresponding 95% confidence interval.

Since patients were recruited before the release of the WHO 2021 guidelines and there does not appear to be any IDH testing for enrollment criteria, it would be helpful to specify whether any IDH-MT patients were recruited, how many, and whether they received treatment.

Why are survival estimates not calculated from the day of diagnosis (i.e., surgery)? This method would ease comparisons with existing literature.

When were Cox models utilized, and were the underlying assumptions verified?

Reviewer #3

(Remarks to the Author)

The manuscript is well-written, and the figures are presented clearly.

The extended arm of the GLORIA trial clearly demonstrates significant improvements in PFS and OS in patients with GBM who did not receive multiple salvage therapy interventions, compared to the dose-escalation phase of the GLORIA trial. The lack of comparison group with RT+BEV would make it difficult to determine the impact of NOX. Overall the study is structured to test toxicity and a possible initial signal of efficacy. The PFS and OS as described, given historical controls, limit enthusiasm for this compound/approach. Additional SOC control groups should be provided that include clinical trials and/or BEV administration. Similarly, to have a radiographic response in the non-Bev group could be problematic given the effect of bev on the MRI. We agree with the authors that an expanded cohort will be necessary to determine ultimate efficacy, but should include a bev control group.

Introduction:

Line 94 – This section lacks sufficient background information on the Hypoxia-VEGF-CXCL12 axis, which is crucial for readers to fully understand the manuscript. The rationale behind VEGF-CXCL12-mediated dual-inhibition therapy should be stated more clearly.

Results:

- Figure 1: I suggest including immunostaining images for each cell type analyzed (endothelial cells, pericytes, TAM, glioma cells), along with their colocalization with CA9 and CXCL12.
- Did the authors examine the correlation between EG12high/low expression and PFS in the RT+NOX-A12+ BEV arm, as was done in the dose-escalation phase of the trial? This should be addressed in the text.

Discussion:

Line 341 – The statement "CXCL12 is a central factor driving pro-tumorigenic remodeling of the TME" is a strong claim. Consider rephrasing this for clarity.

Methods:

For clarity, present the "Buffers and Solutions for Multiplexed Immunofluorescence" section in a table format.

Supplementary Figures:

- Supplementary Figure 3: Please include the title of the figure in the figure legend.
- Perform statistical analysis and report the results in the text.

Version 1:

Reviewer comments:

Reviewer #1

(Remarks to the Author)

The authors have done a really good job in thoughtfully addressing the reviewers comments, providing further translational data supporting their hypotheses. I have no further comments - this is an important area of unmet need.

Reviewer #2

(Remarks to the Author)

My concerns have been addressed

Reviewer #3

(Remarks to the Author)

Concerns adequately addressed

Author's response to reviewer's comments

We thank you and the reviewers for the thoughtful and constructive feedback on our manuscript.

We have carefully considered all comments and revised the manuscript accordingly. As part of this process, particularly extensive time- and resource-consuming spatial tissue analyses of trial and SOC patients were additionally performed to furthermore enhance mechanistic insights as suggested by reviewers #1 and #3.

Below, we provide a detailed, point-by-point response to each reviewer's comment. All changes are highlighted in the revised manuscript, and page/paragraph numbers are provided in our responses for clarity.

We believe these revisions have significantly further improved the manuscript meeting the standards for publication in *Nature Communications*.

Reviewer #1

This is a well-written follow up manuscript of the GLORIA trial reporting a 6 patient safety expansion cohort of CXCL12 inhibition combined with Bevacizumab and radiotherapy in newly diagnosed GBM.

Answer: We thank the reviewer for these positive and encouraging comments regarding the quality and clarity of our manuscript.

The authors provide the rationale of the combination with spatially resolved data obtained from two publically available data sources (TCGA and the Ivy Glioblastoma Atlas Project). This suggests distinct regulatory mechanisms of vascularization in GBM which the authors then explore using multi-colour IF in 5 human GBM samples obtained via SOC surgical procedures. Multi-colour IF does confirm spatially distinct expression of CXCL12 and CA9 (surrogate of hypoxia) supporting their hypothesis.

The authors then report the clinical course of the additional 6 patients treated on the expansion cohort (the escalation data having been previously already published - Giordano 2024. Bevacizumab was given at 10mg/kg -the dose previously used in the Avaglio studies. No additional safety signals were seen, albeit numbers too small to make conclusions. The outcomes of these 6 patients due appear to be better than expected with standard of care, in particular given the poor prognosis patient population enrolled (incomplete resection, PS<2) although I note steroid doses are not reported. No translational data from these 6 patients is actually presented - have the authors got access to these samples as baseline multi-colour IF on these samples would have significantly strengthened the paper.

Answer: We appreciate the reviewer's comment. To strengthen the translational aspect of our work, we have now gained access to the samples and performed extensive further analyses for all the six GLORIA expansion arm patients. The results of these new analyses have been

incorporated into the revised version of the manuscript in Results (page 17) and Discussion (page 18).

We were able to confirm our previous findings for the spatial distribution of CXCL12 (Fig. 1 e, f) also in these samples. The previously established EG12 score, which quantifies CXCL12 expression in the endothelial and glioma cell compartment, was not associated with response in patients receiving combined CXCL12 and VEGF inhibition, in contrast to those treated with CXCL12 inhibition only (as shown previously in Giordano *et al.*, Nat Comm 2024). Collectively, these findings support our model that VEGF and CXCL12 pathways cooperate in revascularization, yet remain functionally and spatially distinct. Thus, predictive biomarkers for CXCL12 inhibition may not be applicable to a combination therapy. However, we acknowledge that the number of patients in the combination arm is low and limits inference. Accordingly, we present these data as exploratory and hypothesis-generating and have included the analyses as a new Supplementary figure (Suppl. Fig. 10).

Results, page 17, paragraph 1: *We next assessed whether outcomes of patients receiving RT+NOX-A12+BEV correlated with CXCL12 expression. A higher frequency of CXCL12+ endothelial and glioma cells (EG12) may serve as a predictive marker for CXCL12-targeted therapy, as it was significantly associated with prolonged PFS under NOX-A12, but not in a SOC cohort.³⁸ In the dose-escalation part of the trial with patients receiving RT+NOX-A12, EG12 correlated significantly with PFS ($r_s = 0.865$, $p = 0.005$). This correlation was lost in patients receiving RT+NOX-A12+BEV (Suppl. Fig. 10), with a negative trend between EG12 and both PFS ($r_s = -0.714$, $p = 0.136$) and OS ($r_s = -0.371$, $p = 0.497$).*

Discussion, page 18, paragraph 3: *We also detected physiological CXCL12 expression in healthy human brain vessels, in line with an emerging role of the CXCL12-CXCR4 axis for early-response to cerebral ischemia in conditions like cerebral insults, as shown in mouse models.^{47,48} It appears likely that comparable damage and repair mechanisms are activated in GBM, where angiogenesis and vasculogenesis appear to be simultaneous, yet spatially distinct and likely complementary mechanisms. This is supported by our findings in publicly available GBM datasets and multi-omics analyses of representative GBM patients, as well as by the translational assessment of tissue from GLORIA patients. The frequency of CXCL12 expression in endothelial and glioma cells was prognostic for therapeutic benefit exclusively in patients receiving CXCL12-targeted therapy only. This association was no longer observed in patients who also received VEGF-targeted treatment, supporting the concept of cooperative, but functionally and spatially distinct signaling between the CXCL12 and VEGF pathways, suggesting that predictive biomarkers for CXCL12 inhibition may not be applicable to a combination therapy.*

Supplementary Figures, page 45: A new Suppl. Fig. 10 was added, showing the correlation of oncologic outcomes with compartmental CXCL12 expression.

Supplementary Figures, page 47: A new Suppl. Fig. 12 was added, providing side-by-side illustration of tumor areas analyzed in multiplexed immunofluorescence staining and corresponding H&E staining of GLORIA expansion arm A patients.

The outcomes of these 6 patients due appear to be better than expected with standard of care, in particular given the poor prognosis patient population enrolled (incomplete resection, PS<2) although I note steroid doses are not reported.

Answer: Baseline steroid exposure has now been added to the manuscript and incorporated into Fig. 3b (formerly Fig. 2). Three patients received dexamethasone at baseline, defined as time of NOX-A12 initiation. We note, however, that patients in the GLORIA trial started treatment with NOX-A12 after a median of 1.1 months post-surgery; at which point ongoing corticosteroid use is generally uncommon (Mistry, Front Oncol 2023).

Results, page 11, Fig. 3b: Baseline steroid doses were added to the figure for all patients.

In conclusion, the authors provided a solid rationale for the exploration of their combination hypothesis - however I am disappointed that there were no mechanistic attempts to explore this in vitro (for example using brain slice cultures from their SOC surgical samples) which would have provided proof of concept given their spatial IF assays are already established. The preliminary clinical data from 6 patients treated in an expansion cohort is intriguing but too small to draw any firm conclusions.

Answer: In the revised manuscript, we now provide substantial new experimental data that offer mechanistic insights, including studies using GBM-derived organoids (GBOs) cultured on human brain slices, as suggested, and spatial transcriptomics at single-cell resolution (Xenium[®] platform) from the brain slices, normal brain tissue, and GBM specimens (new Fig. 2).

First, GBO cultured on human brain slices demonstrated extensive hypoxic areas with strong activation of hypoxia-driven gene programs including VEGFA at the core, whereas CXCL12 expression was sparse and, when present, colocalized predominantly with few endothelial cells, and, to a lesser extent, pericyte-, macrophage- and glioma-related transcripts (new Fig. 2a,b). Notably, CXCL12 expression was detected in normal brain vasculature of the surrounding, the GBO embedding tissue (new Fig. 2c). Due to inherent limitations of the GBO human brain slice culture system, such as the absence of circulation and the restricted culture duration, it was not feasible to investigate angiogenesis, immune cell recruitment, or the recruitment of endothelial progenitor cells.

Taken together, these findings corroborate our results from our immunofluorescence studies on a mechanistic level and transcript level, showing that the induction of VEGFA and CXCL12 appears to be driven by largely complementary mechanisms, whereby hypoxia is the main driver for VEGFA in GBM but not so much or at all for CXCL12. In contrast, CXCL12 expression is already present in vessels of normal human brain and thus appears to be augmented in the microvascular proliferation zones of GBM as also shown by additional spatial transcriptomics of GBM (new Fig. 2d,e).

We have edited the manuscript as follows:

Results, page 8, paragraph 1: *To further test and validate this hypothesis, we next generated patient-derived glioblastoma organoids (GBOs) from surgical specimens cultured on human brain slices (Fig. 2a) and performed spatial transcriptomics on a representative model. The GBO tumor tissue displayed extensive hypoxic areas with pronounced VEGF expression at the core, whereas CXCL12 expression was sparse and, when present, colocalized predominantly with few endothelial cells, and, to a lesser extent, pericyte-,*

macrophage- and glioma-related transcripts (Fig. 2b). Notably, CXCL12 expression was detected in normal brain vasculature of the surrounding, the GBO embedding tissue (Fig. 2c). To overcome the mechanistical limitations of revascularization assessment in a non-perfused GBO model, we conducted parallel spatial transcriptomics analyses on a representative GBM patient specimen with well-defined zonal architecture (Fig. 2d). Consistent with our previously reported observations, CXCL12 expression localized to endothelial cells, pericytes, macrophages/microglia and glioma cells. In contrast, VEGF expression strongly correlated with hypoxia-related genes, while CXCL12 showed only a weak association with hypoxia but a robust link to the described cellular compartments (Fig. 2e).

Taken together with our preceding analyses, these findings further supported the spatially and mechanistically distinct involvement of both VEGF- and CXCL12-mediated pathways in GBM vascular remodeling. Consequently, within the GLORIA trial, we evaluated the potential additional clinical benefit of VEGFA-targeting with BEV in combination with NOX-A12.

Results, page 9, Figure 2: A new Figure 2 was introduced presenting the results of the spatial transcriptomics from patient-derived glioblastoma organoid brain slices and freshly resected glioblastoma specimen.

Discussion, page 18, paragraph 3: *We also detected physiological CXCL12 expression in healthy human brain vessels, in line with an emerging role of the CXCL12-CXCR4 axis for early-response to cerebral ischemia in conditions like cerebral insults, as shown in mouse models.^{47,48} It appears likely that comparable damage and repair mechanisms are activated in GBM, where angiogenesis and vasculogenesis appear to be simultaneous, yet spatially distinct and likely complementary mechanisms. This is supported by our findings in publicly available GBM datasets and multi-omics analyses of representative GBM patients, as well as by the translational assessment of tissue from GLORIA patients. The frequency of CXCL12 expression in endothelial and glioma cells was associated with therapeutic benefit exclusively in patients receiving CXCL12-targeted therapy only. This association was no longer observed in patients who also received VEGF-targeted treatment, supporting the concept of cooperative, but functionally and spatially distinct signaling between the CXCL12 and VEGF pathways, suggesting that potential predictive biomarkers for CXCL12 inhibition may not be applicable to a combination therapy.*

Methods, page 27, 28 and 34: Information was added for the newly introduced techniques.

Reviewer #2:

Overall the manuscript is very well written and quite interesting. I have the following questions:

Answer: We thank the reviewer for the positive assessment of our work and for the thoughtful questions, which we address in detail below.

Are there differences in the clinical and surgical traits between the external contemporary multicentric SOC cohort and the internal histology-matched SOC?

Answer: Ensuring optimal comparability between our trial cohort and the internal and external SOC cohorts was of highest importance.

The internal SOC cohort included patients treated at the neuro-oncological center of the University Hospital Bonn, Germany between 2010 and 2023. Histological criteria for selection of reference patients were: newly-diagnosed GBM, IDH-wildtype (CNS WHO grade 4), absence of *MGMT* promoter methylation. Clinical criteria were: ECOG of 0-2, status post biopsy or incomplete resection, and first line therapy with RT (and optionally TMZ).

The external SOC cohort was derived from a publicly available multicenter dataset with comparable patient characteristics (Drexler *et al.*, 2024). Histological criteria were: newly-diagnosed GBM, IDH-wildtype, absence of *MGMT* promoter methylation. Clinical criteria were: ECOG of 0-2, status post incomplete resection (<90% removal), and first line therapy with RT (and optionally TMZ).

Both cohorts thus represent contemporary patient populations from German neuro-oncological centers treated with SOC, with only minor differences summarized in Extended Data Table 2 and 3 (primarily slight variations in RT fractionation and adjuvant temozolomide administration). For full transparency, the patient characteristics of both cohorts are provided in the Source Data File. The Methods section was revised accordingly to clarify cohort comparability.

Methods, page 23, paragraph 2: *We benchmarked oncological outcomes of the GLORIA trial to an internal (University Hospital Bonn)³⁸ and a recently published external German multicenter cohort⁴⁴ of GBM patients with comparable patient characteristics resembling inclusion criteria of the GLORIA trial treated with standard RT and optional TMZ.*

Methods, page 23, paragraph 2: *The patient characteristics of both cohorts are provided in Extended Data Table 2 and 3 and the Source Data File.*

Why was progression-free survival (PFS) data only available for 39 out of 64 patients in the external cohort? Could this skew the results, making the median PFS appear higher?

Answer: We cannot comment on why PFS data were available for only a subset of patients in the external SOC cohort, as none of our authors were involved in that analysis. However, loss to follow-up causing PFS censoring is common in oncological standard-of-care cohorts treated in a routine setting. Given that said data set has been published very recently (Drexler *et al.*, Nat Med, 2024) we consider the data to be robust and representative of current SOC

outcomes for this GBM patient subgroup. PFS is well-known to be influenced by the number of informatively censored patients. We acknowledge that the high frequency of PFS censoring may have introduced bias, potentially leading to an overestimation of the true PFS in this cohort. However, this would rather *disadvantage* the GLORIA cohort in direct comparison, and we have now addressed this potential effect in the Discussion.

Discussion, page 18, paragraph 2: *Both PFS and OS in GLORIA patients receiving NOX-A12 + BEV appear favorable to comparable contemporary external SOC cohorts. However, the high frequency of PFS censoring in these⁴⁴ may have introduced bias, potentially leading to an overestimation of the true PFS, thereby disadvantaging the GLORIA cohort in direct comparison.*

It appears that there is a significant confounder, as patients in Arm B of the 6-arm expansion may have undergone a gross total resection (GTR). Why was this permitted?

Answer In the initial trial protocol, only patients with residual tumor were eligible for the trial. Following a change of the protocol under active trial recruitment, GTR was then permitted in all arms of the trial. This decision was made as there was no obvious reason to question the drugs' mode of action in patients lacking MRI contrast-enhancing tumor and the sponsor wished to further explore the treatment also in this clinical setting.

However, none of the patients that eventually were recruited, had undergone GTR. All GLORIA patients in all arms had an incomplete resection or were not amenable to resection. Of note, arm B of the expansion was never recruited, as the sponsor decided to focus on the combinatory treatment of RT+NOX-A12+BEV after considering the first interim results in arm A.

We added a statement in the Results section to clarify this aspect.

Results, page 10, paragraph 3: *No patient had undergone gross total resection.*

What was the approach to censoring patient C2-001?

Answer: According to the pre-defined statistical analysis plan and after careful consideration with the statistician and the sponsor, C2-001 required censoring for PFS due to the individual patient course (see below). To ensure full transparency, we exceed journal requirements and decided to provide patient narratives as a Supplementary Note for all trial patients:

Supplementary Note 1 (patient narratives), page 2, paragraph 2: *This is a patient with a frontotemporal GBM. The patient underwent partial resection on May 22, 2020. After proper diagnosis, the patient was included in GLORIA. The patient was started on 400 mg/week NOX-A12 on June 22, 2020 and was treated with hypofractionated RT. The first scan in week 9 showed stable disease. The scan performed on October 19, 2020 (week 18) suggested tumor recurrence (pPD) which was not confirmed in a further scan. Of note, at the time point of this MRI no AEs were reported and the time point NANO score remained 0. Despite this, the PI decided to discontinue NOX-A12 and initiate salvage therapy with temozolomide on November 18, 2020 without performing an MRI scan. On January 6, 2021 the patient received an infusion of bevacizumab. Although a scan performed 3 weeks later showed PR, while previous pseudo-*

progression cannot be excluded, we considered the overall time point response as “not evaluable” due to “bevacizumab blinding”. (...)

Since salvage therapy was i) initiated in the absence of confirmed PD and ii) no clinical deterioration was noted at the previous scan date, the patient is censored for PFS at the time point of the scan preceding the initiation of salvage therapy (October 19, 2020).

Final interpretation:

- *PD as per RANO is indeterminable, resulting in PFS being censored at 120 days*
- *The overall survival time was 287 days*

It appears that medians and ranges were calculated using an Excel spreadsheet. Was this the method for conducting the survival analysis?

Answer: No calculations and statistical analyses were conducted using Microsoft Excel. The provided Excel spreadsheet is for information of the reader only as an obligatory submission requirement by the Editorial Office of Nature Communications to assure access to the raw data for the public.

All statistical tests, including descriptive statistics, were performed using GraphPad Prism 10 and R version 4.2.2 as specified in the figure legends and the Methods section.

It's unusual to report ranges for median survival; generally, the median from a Kaplan-Meier estimate should include the corresponding 95% confidence interval.

Answer: After consulting with our in-house statistician, we initially decided to provide ranges instead of 95% confidence intervals to avoid potentially misleading associations given the overall number of patients in the expansion arm is only six individuals. We considered the combination of ranges and median values to best reflect the data's distribution in this relatively small patient collective. Following the reviewer's recommendation, we have now added 95% confidence intervals. Please note that, due to the small number of events, the upper confidence limit could not be estimated in some cases, this is then clearly indicated in the text.

Results, page 15, paragraph 2 and page 16, paragraph 2: 95% confidence intervals have been added for survival outcomes.

Since patients were recruited before the release of the WHO 2021 guidelines and there does not appear to be any IDH testing for enrollment criteria, it would be helpful to specify whether any IDH-MT patients were recruited, how many, and whether they received treatment.

Answer: All patients were confirmed as glioblastoma, IDH-wildtype (CNS WHO grade 4) according to the WHO 2021 classification for CNS tumors by immunohistochemistry. If patients were younger than 54 years, they were assessed additionally by pyrosequencing for IDH1 and IDH2.

To clarify, we made the following modification to the Results and Methods sections of the manuscript:

Results, page 10, paragraph 2: *The GLORIA trial (NCT04121455) is a multicentric phase 1/2 trial conducted at six major neuro-oncology centers in Germany. It was designed to*

evaluate the safety and efficacy of RT combined with continuous intravenous (i.v.) treatment with NOX-A12 in newly-diagnosed, incompletely resected IDH-wildtype GBM (CNS WHO grade 4) lacking MGMT promoter methylation (Fig. 3a).

Results, page 11, Fig. 3b: IDH wildtype status was added to the figure for all patients.

Methods, page 21, paragraph 1: *All patients were neuropathologically confirmed as GBM, IDH-wildtype (CNS WHO grade 4) according to the 2021 WHO classification for CNS tumors. If patients were ≤ 54 years of age, IDH mutations were additionally assessed by pyrosequencing of the IDH1 and IDH2 loci.*

Why are survival estimates not calculated from the day of diagnosis (i.e., surgery)? This method would ease comparisons with existing literature.

Answer: To enhance comparability, we have now additionally provided survival estimates calculated from the date of initial surgery.

We agree that most trials rather report survival estimates calculated from the day of surgery. However, several recent trials, such as EF14 (Stupp et al., 2017), defined survival from the initiation of experimental treatment. In accordance with the predefined trial protocol, survival in our analysis was calculated from the start of experimental treatment with the CXCL12 inhibitor. We acknowledge that this approach is indeed disadvantageous for the GLORIA cohorts and this limitation is now emphasized more prominently in the Discussion. Please note that the *per-protocol* analysis remains unchanged, as it was predefined in the trial protocol as a secondary endpoint.

Results, page 16, paragraph 3: *To enhance comparability with the SOC cohorts and existing literature, we additionally calculated GLORIA survival outcomes from the date of initial surgery. Both PFS and OS of patients receiving NOX-A12+BEV were approximately 1.1 months longer, without affecting the statistical significance of comparisons with the SOC cohorts (Suppl. Fig. 8).*

Supplementary Figures, page 43: A new Suppl. Fig. 8 was added, showing GLORIA survival outcomes from the date of initial surgery in comparison to the external and the internal histology-matched SOC cohort.

Supplementary Figures, page 44: Suppl. Fig. 9 (previously 7) was modified, now additionally showing GLORIA survival outcomes from the date of initial surgery in comparison to SOC and literature cohorts.

Discussion, page 18, paragraph 2: *In contrast, many studies initiate treatment earlier or commonly define the date of surgery as starting point for outcome assessment. Consequently, the per protocol GLORIA survival data should be interpreted with caution, as it likely underestimates survival compared with most contemporary trials.*

When were Cox models utilized, and were the underlying assumptions verified?

Answer: The primary outcome analysis was performed using the Kaplan–Meier method and log-rank tests, which do not rely on proportional hazards assumptions. We agree that the additional Cox proportional hazards regression analyses have limited interpretability given the small sample size in this trial. However, hazard ratios were included for completeness and to facilitate comparison with other clinical reports, in line with common reporting standards. The proportional hazard assumption was evaluated graphically using log-log survival plots and Schoenfeld residuals and appeared reasonable within the limitations of the dataset. The corresponding Methods section was revised accordingly, and the Discussion expanded to further emphasize the limited statistical power of the survival analyses.

Discussion, page 19, paragraph 2: *While the small sample size limits the statistical interpretability and power to evaluate the survival impact of compartmental CXCL12 expression within this subset, the results still provide important hypothesis-generating insights.*

Methods, page 28, paragraph 1: *Survival rates were estimated using the Kaplan–Meier method and compared by log-rank test. Exploratory Cox proportional hazards regression analyses were additionally performed, and the proportional hazards assumption was verified using log-log survival plots and Schoenfeld residuals.*

Reviewer #3:

The manuscript is well-written, and the figures are presented clearly.

Answer: We thank the reviewer for this positive assessment of our work and for the thoughtful questions. Please find our remarks below.

The extended arm of the GLORIA trial clearly demonstrates significant improvements in PFS and OS in patients with GBM who did not receive multiple salvage therapy interventions, compared to the dose-escalation phase of the GLORIA trial. The lack of comparison group with RT+BEV would make it difficult to determine the impact of NOX. Overall the study is structured to test toxicity and a possible initial signal of efficacy. The PFS and OS as described, given historical controls, limit enthusiasm for this compound/approach. Additional SOC control groups should be provided that include clinical trials and/or BEV administration. Similarly, to have a radiographic response in the non-Bev group could be problematic given the effect of bev on the MRI. We agree with the authors that an expanded cohort will be necessary to determine ultimate efficacy, but should include a bev control group.

Answer: We agree that the triple treatment in the GLORIA expansion arm with addition of BEV clearly improved outcomes at unchanged good tolerability. Indeed, ultimate interpretations of efficacy requires a randomized control group which is planned for the upcoming phase 2/3 trial currently in preparation. The characteristics of the control groups will be ultimately determined by the official FDA requests. Unfortunately, SOC patients treated with BEV in the first line are not available in Europe, as BEV was never approved for GBM by the EMA. Therefore, historical high-quality outcome data for BEV only are available from the two phase 3 trials by Gilbert *et al.* and Chinot *et al.* which are referenced and discussed in the manuscript. Additionally, Ulrich Herrlinger, co-author (co-senior) of this manuscript, has shared his extensive expertise on BEV treatment as former PI of the GLARIUS trial. Carefully taking the obvious limitation of a phase 1/2 trial into consideration, we believe that the survival outcome for RT+NOX-A12+BEV warrants further assessment in a follow-up trial on efficacy. Direct comparison clearly requires caution, but the outcome of GLORIA expansion arm A exceeds our SOC cohorts as well as historical controls with comparable patient characteristics. An overview and comparison of these cohorts is provided in the revised Suppl. Fig. 9. We also revised the manuscript following this reviewer comment to tone down our statements and strengthen said aspects in the Discussion. Notably, in GLORIA, survival endpoints were not calculated from day of diagnosis (as is common in most of the historical SOC and BEV cohorts), but from the day of initiation of experimental treatment, which is around 1.1 months later. Consequently, the per protocol GLORIA survival data likely underestimates survival compared with most contemporary trials.

Our trial results also emphasize the importance of advanced MRI analyses to prevent misinterpretation of radiographic pseudoresponse and pseudoprogression. We expanded the Discussion section to address the points raised by the reviewer.

Discussion, page 18, paragraph 2: *While the PFS observed in GLORIA patients receiving BEV is consistent with outcomes reported in recent (larger-scaled) clinical trials of the VEGFA-targeting drug,²⁰⁻²² the combinatory approach resulted in a median OS of 19.9 months from NOX-A12 initiation and 2/6 patients surpassing 2 year-OS, suggesting a trend toward*

improved long-term outcomes. Direct comparisons remain challenging, as the referenced trials enrolled overall more prognostically favorable patient populations. We consider the here reported GLORIA results a robust OS signal, especially since we only included GBM patients with highly unfavorable prognosis.

Discussion, page 19, paragraph 2: *In a future phase 2/3 trial, particular emphasis should be placed on the inclusion of adequate control arms and rigorous radiographic response assessment. Given the vascular mechanism of the therapeutic approach presented here, advanced MRI techniques will be essential to reliably distinguish true response from true progression.*

Introduction: Line 94 – This section lacks sufficient background information on the Hypoxia-VEGF-CXCL12 axis, which is crucial for readers to fully understand the manuscript. The rationale behind VEGF-CXCL12-mediated dual-inhibition therapy should be stated more clearly.

Answer: We have expanded the Introduction to include additional background on the Hypoxia–CXCL12 signaling axis. We furthermore clarified the rationale for VEGF–CXCL12 dual inhibition in GBM.

Introduction, page 4, paragraph 4: *Radiation-induced loss of endothelial cells, accumulation of reactive oxygen species, and aggressive tumor growth with aberrant neovascularization collectively lead to impaired tumor perfusion triggering TGF- β -augmented induction of the hypoxia-related factor HIF1 α . This, in turn, attracts CXCR4+ BMDCs along a gradient of the CXCR4 ligand CXCL12.^{24,28,29} Elevated CXCL12 expression facilitates not only BMDC migration but also their adhesion and homing to hypoxic tumor regions.²⁴*

Introduction, page 5, paragraph 1: *To date, no clinical trial has investigated dual targeting of CXCL12 and VEGF as two key mediators of post-radiogenic revascularization in GBM.*

Results: Figure 1: I suggest including immunostaining images for each cell type analyzed (endothelial cells, pericytes, TAM, glioma cells), along with their colocalization with CA9 and CXCL12.

Answer: We have now included representative immunostaining images for endothelial cells, pericytes, TAMs, and glioma cells, showing respective colocalization with CA9 and CXCL12 in Figure 1d. Additional images are provided in a new Suppl. Fig. 2.

Results, page 6, paragraph 3: *Aiming for single-cell spatial resolution, we performed multiplex-immunofluorescence (mIF) stainings on human GBM tissue samples (n = 5, Fig. 1d and Suppl. Fig. 2).*

Results, page 7, Figure 1: Immunostaining images for each cell type analyzed and their colocalization with CA9 and CXCL12 have been added in Fig. 1d.

Supplementary Figures, page 36: A new Suppl. Fig. 2 was additionally provided, showing more representative immunostaining images for all the cell types and respective colocalization with CA9 and CXCL12.

Did the authors examine the correlation between EG12high/low expression and PFS in the RT+NOX-A12+ BEV arm, as was done in the dose-escalation phase of the trial? This should be addressed in the text.

Answer: Following the reviewers' comment, we have gained access to the tissue of all expansion arm patients and now performed the requested analysis correlating EG12 expression levels with survival outcomes in the RT+NOX-A12+BEV arm (see also Reviewer #1). The results are included in the revised manuscript and discussed accordingly.

We were able to confirm our previous findings for the spatial distribution of CXCL12 also in the expansion arm patient samples. The previously established EG12 score, which quantifies CXCL12 expression in the endothelial and glioma cell compartment, was not associated with response in patients receiving combined CXCL12 and VEGF inhibition, in contrast to those treated with CXCL12 inhibition only (as shown previously in Giordano *et al.*, 2024). These findings support our concept that VEGF and CXCL12 pathways cooperate in revascularization but are also functionally and spatially distinct. Thus, predictive biomarkers for CXCL12 inhibition may not be applicable to a combination therapy. However, we are also aware that the number of patients is low and thus scientific value remains limited as also pointed out in the manuscript. We added the analyses in an additional Supplementary figure.

Results, page 17, paragraph 1: *We next assessed whether outcomes of patients receiving RT+NOX-A12+BEV correlated with CXCL12 expression. A higher frequency of CXCL12+ endothelial and glioma cells (EG12) may serve as a predictive marker for CXCL12-targeted therapy, as it was significantly associated with prolonged PFS under NOX-A12, but not in a SOC cohort.³⁸ In the dose-escalation part of the trial with patients receiving RT+NOX-A12, EG12 correlated significantly with PFS ($r_s = 0.865$, $p = 0.005$). This correlation was lost in patients receiving RT+NOX-A12+BEV (Suppl. Fig. 10), with a negative trend between EG12 and both PFS ($r_s = -0.714$, $p = 0.136$) and OS ($r_s = -0.371$, $p = 0.497$).*

Discussion, page 18, paragraph 3: *We also detected physiological CXCL12 expression in healthy human brain vessels, in line with an emerging role of the CXCL12-CXCR4 axis for early-response to cerebral ischemia in conditions like cerebral insults, as shown in mouse models.^{47,48} It appears likely that comparable damage and repair mechanisms are activated in GBM, where angiogenesis and vasculogenesis appear to be simultaneous, yet spatially distinct and likely complementary mechanisms. This is supported by our findings in publicly available GBM datasets and multi-omics analyses of representative GBM patients, as well as by the translational assessment of tissue from GLORIA patients. The frequency of CXCL12 expression in endothelial and glioma cells was associated with therapeutic benefit exclusively in patients receiving CXCL12-targeted therapy only. This association was no longer observed in patients who also received VEGF-targeted treatment, supporting the concept of cooperative, but functionally and spatially distinct signaling between the CXCL12 and VEGF pathways, suggesting that potential predictive biomarkers for CXCL12 inhibition may not be applicable to a combination therapy.*

Supplementary Figures, page 45: A new Suppl. Fig. 10 was added, showing the correlation of oncologic outcomes with compartmental CXCL12 expression in the respective GLORIA cohorts.

Supplementary Figures, page 47: A new Suppl. Fig. 12 was added, providing side-by-side illustration of tumor areas analyzed in multiplexed immunofluorescence staining and corresponding H&E staining of GLORIA expansion arm A.

Discussion: Line 341 – The statement "CXCL12 is a central factor driving pro-tumorigenic remodeling of the TME" is a strong claim. Consider rephrasing this for clarity.

Answer: We have rephrased and toned down this statement for clarity.

Discussion, page 19, paragraph 1: *Our findings support this hypothesis, suggesting that CXCL12 contributes substantially to pro-tumorigenic remodeling of the TME and, ultimately, GBM recurrence after RT.*

Methods: For clarity, present the "Buffers and Solutions for Multiplexed Immunofluorescence" section in a table format.

Answer: We have reformatted this section into a table to improve clarity and readability.

Methods, page 26: *Table 1: Buffers and solutions for multiplexed immunofluorescence*

Supplementary Figures: Supplementary Figure 3: Please include the title of the figure in the figure legend.

Answer: We have now added the title to the legend of Supplementary Figure 4 (previously Suppl. Fig. 3) as requested.

Perform statistical analysis and report the results in the text.

Answer: Statistical analyses have been performed for the relevant comparisons and added to the Methods. The corresponding results are now reported in the Results section and Supplementary Fig. 4 (previously Suppl. Fig. 3).

Results, page 12, paragraph 1: *With increasing NOX-A12 dosage, plasma levels increased correspondingly ($p = 0.007$), exceeding the stable CXCL12 levels ($p = 0.116$; Suppl. Fig. 4). NOX-A12 plasma levels were significantly higher from DL 1 to DL 2 ($p = 0.020$) and DL3 ($p = 0.020$), but did not differ significantly with additional BEV treatment ($p = 0.270$).*

Methods, page 28, paragraph 4: *Group comparisons of drug and target concentrations at identical time points were performed using the Kruskal–Wallis test, followed by pairwise Mann–Whitney tests. For integrated pharmacokinetic evaluation, the area under the curve (AUC) was calculated for each patient. Between-group differences in AUCs were assessed using Welch ANOVA.*

Supplementary Figures, page 39: Suppl. Fig. 4 (previously Suppl. Fig. 3) was modified, now additionally providing statistical analyses.